# Dilated cardiomyopathy mutation E525K in human beta-cardiac myosin stabilizes the interacting-heads motif and super-relaxed state of myosin

David V Rasicci[1], Prince Tiwari[2], Skylar ML Bodt[1], Rohini Desetty[1], Fredrik R Sadler[3], Sivaraj Sivaramakrishnan[3], Roger Craig[2], Christopher M Yengo[1]*

[1]Department of Cellular and Molecular Physiology, Penn State College of Medicine, Hershey, United States; [2]Department of Radiology, Division of Cell Biology and Imaging, UMass Chan Medical School, Worcester, United States; [3]Department of Genetics, Cell Biology, and Development, University of Minnesota Twin Cities, Minneapolis, United States

*For correspondence:
cmy11@psu.edu

**Abstract** The auto-inhibited, super-relaxed (SRX) state of cardiac myosin is thought to be crucial for regulating contraction, relaxation, and energy conservation in the heart. We used single ATP turnover experiments to demonstrate that a dilated cardiomyopathy (DCM) mutation (E525K) in human beta-cardiac myosin increases the fraction of myosin heads in the SRX state (with slow ATP turnover), especially in physiological ionic strength conditions. We also utilized FRET between a C-terminal GFP tag on the myosin tail and Cy3ATP bound to the active site of the motor domain to estimate the fraction of heads in the closed, interacting-heads motif (IHM); we found a strong correlation between the IHM and SRX state. Negative stain electron microscopy and 2D class averaging of the construct demonstrated that the E525K mutation increased the fraction of molecules adopting the IHM. Overall, our results demonstrate that the E525K DCM mutation may reduce muscle force and power by stabilizing the auto-inhibited SRX state. Our studies also provide direct evidence for a correlation between the SRX biochemical state and the IHM structural state in cardiac muscle myosin. Furthermore, the E525 residue may be implicated in crucial electrostatic interactions that modulate this conserved, auto-inhibited conformation of myosin.

## Editor's evaluation

This fundamental study demonstrates that a point mutation resulting in dilated cardiomyopathy in human cardiac myosin increases the fraction of molecules that adopt the auto-inhibited super-relaxed conformation. This provides a mechanism for the lower force output observed in the hearts of affected individuals. The data supporting this, utilizing kinetic methods, a FRET-biosensor to detect conformational changes, and electron microscopy are convincing.

## Introduction

Muscle contraction is driven by the sliding of thick and thin filaments in the muscle sarcomere. In striated muscle, contraction is regulated by both thin and thick filament mechanisms. It is well established that a rise in intracellular calcium concentration changes the conformation of the actin-containing thin filaments (thin filament regulation), allowing myosin heads in the thick filament to bind to actin and power filament sliding (*Kobayashi and Solaro, 2005*). Recently, thick filament regulation has

garnered much attention, as myosin heads can form an auto-inhibited or super-relaxed (SRX) state with slow ATP turnover (*Hooijman et al., 2011*; *Nag and Trivedi, 2021*; *Stewart et al., 2010*). Structural studies have shown that myosin heads in the thick filament can fold back on the myosin tail and interact with each other in a conformation referred to as the interacting-heads motif (IHM; *Alamo et al., 2018*; *Woodhead et al., 2005*; *Zoghbi et al., 2008*), which prevents them from interacting with actin. However, it is currently unclear if the SRX biochemical state and IHM structural state are directly correlated (*Craig and Padrón, 2022*).

The SRX state is proposed to play several important roles in cardiac muscle. It appears to play a crucial role in conserving energy, as SRX heads turn over ATP about 5–10 times slower than the uninhibited disordered relaxed (DRX) heads (*Hooijman et al., 2011*; *Toepfer et al., 2020*). Cardiac muscle may also use the SRX heads as a reserve that can be recruited when peripheral metabolic demands warrant-increased cardiac contractility (*McNamara et al., 2015*). It has been hypothesized that regulation of the SRX state in cardiac muscle underlies the Frank-Starling mechanism, in which contractile force increases as cardiac muscle is stretched (i.e. length-dependent activation; *Campbell et al., 2018*; *Kampourakis and Irving, 2021*; *Zhang et al., 2017*).

Point mutations in cardiac myosin can lead to various forms of cardiomyopathy, with hypertrophic cardiomyopathy (HCM) and dilated cardiomyopathy (DCM) being the most common phenotypes (*Yotti et al., 2019*). HCM most commonly presents clinically as hypertrophy of the interventricular septum, while other forms of hypertrophy are also found including concentric, midventricular, and apical. The hypertrophy results in decreased left ventricular chamber size and is accompanied by pronounced relaxation deficits, myofilament disarray, and fibrosis. Preclinical individuals have been found to display increased ejection fraction and impaired relaxation suggesting a hypercontractile phenotype occurs prior to hypertrophy (*Ho et al., 2002*; *Nagueh et al., 2003*). Recent work has revealed that several HCM mutations in cardiac myosin can destabilize the SRX state (*Adhikari et al., 2019*; *Anderson et al., 2018*; *Gollapudi et al., 2021*; *Sarkar et al., 2020*; *Vander Roest et al., 2021*), which may explain the impaired relaxation and hypercontractile phenotype. On the other hand, DCM presents as an increase in left ventricular chamber size which leads to thinning of the myocardium, cardiomyocyte cell death, and a hypocontractile phenotype (*McNally and Mestroni, 2017*). Interestingly, early-stage DCM patients present with subtle decreases in contractility and ejection fraction, suggesting hypocontractility may trigger disease onset (*Lakdawala et al., 2012b*; *Schafer et al., 2017*). An attractive hypothesis is that DCM mutations may stabilize the SRX state, which reduces the number of myosin heads available to produce contractile force and could explain the observed hypocontractility. However, the leading hypothesis is that DCM mutations reduce the intrinsic motor properties of myosin without altering the SRX state (*Robert-Paganin et al., 2018*; *Ujfalusi et al., 2018*), although few studies have examined the impact of DCM mutations on the SRX state (*Yuan et al., 2022*).

The IHM, which appears to be a conserved regulatory structure in all muscle types (*Alamo et al., 2018*; *Lee et al., 2018*), has been proposed to be the structural basis of the SRX state (*Craig and Padrón, 2022*). The motif is found in both single myosin molecules and native thick filaments. Early studies with smooth and non-muscle myosin revealed a folded tail conformation that was proposed to be auto-inhibitory (*Craig et al., 1983*; *Onishi and Wakabayashi, 1982*; *Trybus et al., 1982*; *Trybus and Lowey, 1984*). A study with smooth muscle myosin was the first to reveal the interactions between the heads that characterize the IHM (*Wendt et al., 1999*; *Wendt et al., 2001*) and later how the folded tail stabilizes the interacting heads (*Burgess et al., 2007*). The motif subsequently has been demonstrated in myosin II molecules across the evolutionary tree, from humans to the earliest animals (e.g. sponges; *Jung et al., 2008*; *Lee et al., 2018*). Further biochemical studies demonstrated that the IHM is the auto-inhibited structure that predominates in the dephosphory-lated off-state (*Burgess et al., 2007*; *Cross et al., 1988*; *Wendt et al., 2001*). Three recent cryo-EM studies of smooth muscle myosin solved the high-resolution structure of the IHM, providing crucial details about head-head and head-tail interactions that stabilize the structure (*Heissler et al., 2021*; *Scarff et al., 2020*; *Yang et al., 2020*). The IHM was first demonstrated in native thick filaments in tarantula muscle (*Woodhead et al., 2005*) and then was shown to be conserved in thick filaments throughout the animal kingdom, including vertebrates (*AlKhayat et al., 2013*; *Alamo et al., 2017*; *Zoghbi et al., 2008*). The highest resolution so far (13 Å) has come from tarantula filaments (*Yang et al., 2016*), which showed an IHM with similar features to smooth muscle myosin (*Alamo et al.,*

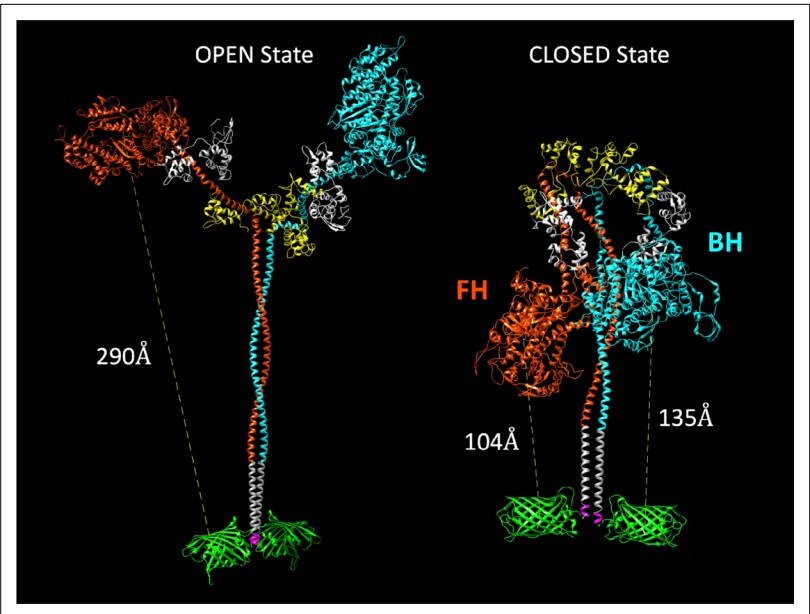

**Figure 1.** Molecular models of the M2β 15HPZ.GFP construct in open and closed (interacting-heads motif [IHM]) states. From proximal to distal, the model contains myosin heavy chains (orange and cyan), each with one essential light chain (white) and one regulatory light chain (yellow). Following the last residue of the 15 heptad tail (E946), the leucine zipper (PDB: 1ZIL, gray) helps to dimerize the construct and is followed by a short, flexible glycine linker (pink), and then the C-terminal GFP molecule (PDB: 1GFL, green). Distances in the figure are measured from Y66 within the tripeptide fluorophore of GFP to S180 in the ATP-binding site of the myosin motor domain. Models were constructed by joining existing pdb files in UCSF Chimera software. The open state is based on a previously published model by the Spudich lab (*Trivedi et al., 2018*), while the closed state is based on the 5TBY cardiac homology model from the Padrón lab (*Alamo et al., 2017*). In the closed state, the free head (FH) of myosin lies closer to the C-terminal GFP molecules, as compared to the blocked head (BH).

*2018*; *Alamo et al., 2008*). However, in skeletal and cardiac myosin, the high-resolution structure of the IHM has not been solved.

In order to directly examine the correlation between the IHM and SRX state, a method that can detect the IHM in solution in addition to the fraction of molecules with slow ATP turnover is crucial. In the current study, we developed a method of detecting the IHM in solution by FRET, allowing us to directly correlate the fraction of myosin heads in the IHM with the fraction of myosin heads in the slow ATP turnover (SRX) state. In addition, we examined the impact of a DCM-associated mutation, E525K, on the formation of the IHM structural and SRX biochemical states. We found that the E525K mutation dramatically stabilizes the SRX state, monitored by single ATP turnover, as well as the IHM, monitored by FRET and confirmed by single molecule EM imaging. Our results suggest that stabilization of the off-state is a viable explanation for the decrease in muscle force and power in DCM. We also demonstrate that the IHM correlates well with the SRX state, providing evidence that the IHM is the structural basis of the SRX biochemical state.

## Results

### Overall approach and rationale

Our initial goal was to produce a human beta-cardiac myosin construct that would allow us to monitor formation of the SRX biochemical and IHM structural states in solution. Previous studies in smooth muscle myosin demonstrated that at least 15 heptads of the S2 region of myosin were required for phosphorylation-dependent regulation (*Trybus et al., 1997*). Thus, we generated human beta-cardiac heavy meromyosin (HMM) with 15 heptads of S2, a leucine zipper, and a C-terminal GFP tag (M2β 15HPZ.GFP; *Figure 1*). We expressed and purified M2β 15HPZ.GFP using the C2C12 expression system (*Chow et al., 2002*; *Wang et al., 2003*), and the myosin contained endogenous mouse skeletal muscle light chains. This is similar in composition to our published studies on S1 (*Rasicci et al.,*

*2021*; *Swenson et al., 2017*; *Tang et al., 2021*). We reasoned that if M2β 15HPZ.GFP forms the IHM then we should be able to observe FRET between Cy3ATP bound to the active site and the C-terminal GFP tag, thus essentially functioning as a biosensor of the IHM or closed state. We generated a model of the open and closed states of M2β 15HPZ.GFP and found that the predicted distance between the active site of the free head (FH) and its C-terminal GFP tag is about 90–140 Å, assuming the GFP tags are flexibly linked to the tail, which is close enough to participate in FRET. All other donor-acceptor pairs (blocked head [BH] to either C-terminal GFP tag, FH to the BH C-terminal GFP) have distances ~130–140 Å (*Figure 1*). In contrast, the distance in the open state is predicted to be ≥250 Å, which is much too far to participate in FRET. Therefore, we proposed that the FRET biosensor would be able to detect the IHM formation in solution and that the FRET distance observed would be an ensemble average of all donor acceptor pairs, which are in the range of 90–140 Å apart in the IHM.

## Actin-activated ATPase and in vitro motility

The WT and E525K M2β 15HPZ.GFP preparations were >95% pure based on SDS-PAGE and similar in purity to our M2β S1 preparations (*Figure 2—figure supplement 1*). To examine the number of active heads in the M2β S1 and 15HPZ.GFP preparations, we performed high-salt $NH_4^+$ ATPase assays in the absence of actin (*Trybus, 2000*). The ATPase activity per head was similar when comparing WT S1 to WT 15HPZ.GFP (4.24±0.09 and 4.14±0.08 s⁻¹, respectively) and E525K S1 to E525K 15HPZ.GFP (5.03±0.09 and 4.93±0.03 s⁻¹, respectively). However, E525K displayed a 20% higher $NH_4^+$ ATPase rate in both the S1 and 15HPZ.GFP preparations. Our $NH_4^+$ ATPase and SDS-PAGE results demonstrate that the M2β 15HPZ.GFP construct does not disrupt the folding of the myosin motor domain.

In order to examine the ability of M2β 15HPZ.GFP to form the SRX state, we first examined steady-state ATPase activity in the presence of varying concentrations of actin (*Figure 2A*). We predicted that the ATPase activity would be decreased compared to our previous studies with monomeric S1, if the HMM construct could form the SRX state. The maximum actin-activated ATPase activity ($k_{cat}$) of WT M2β 15HPZ.GFP was reduced sevenfold compared to our measurements with WT M2β S1 (*Tang et al., 2021*), while the actin concentration at which the ATPase activity is one-half maximal ($K_{ATPase}$) was not significantly different in low-salt conditions (20 mM KCl; *Table 1*). In the WT and E525K M2β 15HPZ.GFP constructs, the $k_{cat}$ and $K_{ATPase}$ were not significantly different.

In order to determine if the E525K mutation alters the intrinsic ATPase activity, in the absence of regulation by the IHM formation, we examined actin-activated ATPase activity in the WT and E525K M2β S1 constructs in parallel (*Figure 2B*, *Table 1*). To our surprise, we found that the mutation increases $k_{cat}$ 20% and dramatically reduces the $K_{ATPase}$ 30-fold. Thus, the E525K mutation enhances catalytic efficiency in S1, but the SRX state is still able to suppress actin-activated ATPase activity in the double-headed HMM construct.

We also performed in vitro gliding assays and found that the average in vitro gliding velocity was similar in WT and E525K M2β 15HPZ.GFP (*Figure 2C* and *Figure 2—video 1*). The velocities were similar to previous reports with a longer HMM construct (*Winkelmann et al., 2015*) but slower than our published results with S1 (*Tang et al., 2021*). There was a larger number of stuck filaments in our HMM experiments compared to S1 (*Table 1*), suggesting that myosin in the auto-inhibited SRX state may adopt a conformation on the motility surface that can bind actin and generate a drag force that opposes the filament sliding.

## Single mantATP turnover experiments

We used mant-labeled ATP (mant labeled at the 3' ribose position) to monitor single turnover rate constants in the absence of actin (*Figure 3* and *Table 2*). Previous studies have demonstrated that cardiac myosin exhibits two phases in single turnover measurements, a fast phase that is similar to the basal ATPase rate of S1 and a slow phase that is 5–10-fold slower and referred to as the SRX state (*Anderson et al., 2018*; *Hooijman et al., 2011*; *Rohde et al., 2018*). Determining the relative amplitudes of the fast and slow phase rate constants allows determination of the fraction of myosin molecules in the open (DRX) and SRX states, respectively. We performed single turnover measurements at various KCl concentrations and observed that KCl destabilizes the SRX state in WT M2β 15HPZ.GFP, becoming a minor component at 150 mM KCl (*Figure 3A*, C–figure supplement 1). However, E525K exhibited a much higher fraction of molecules in the SRX state compared to WT, especially at higher KCl concentrations (*Figure 3B*, C-figure supplement 2). Thus, our results demonstrate E525K stabilizes

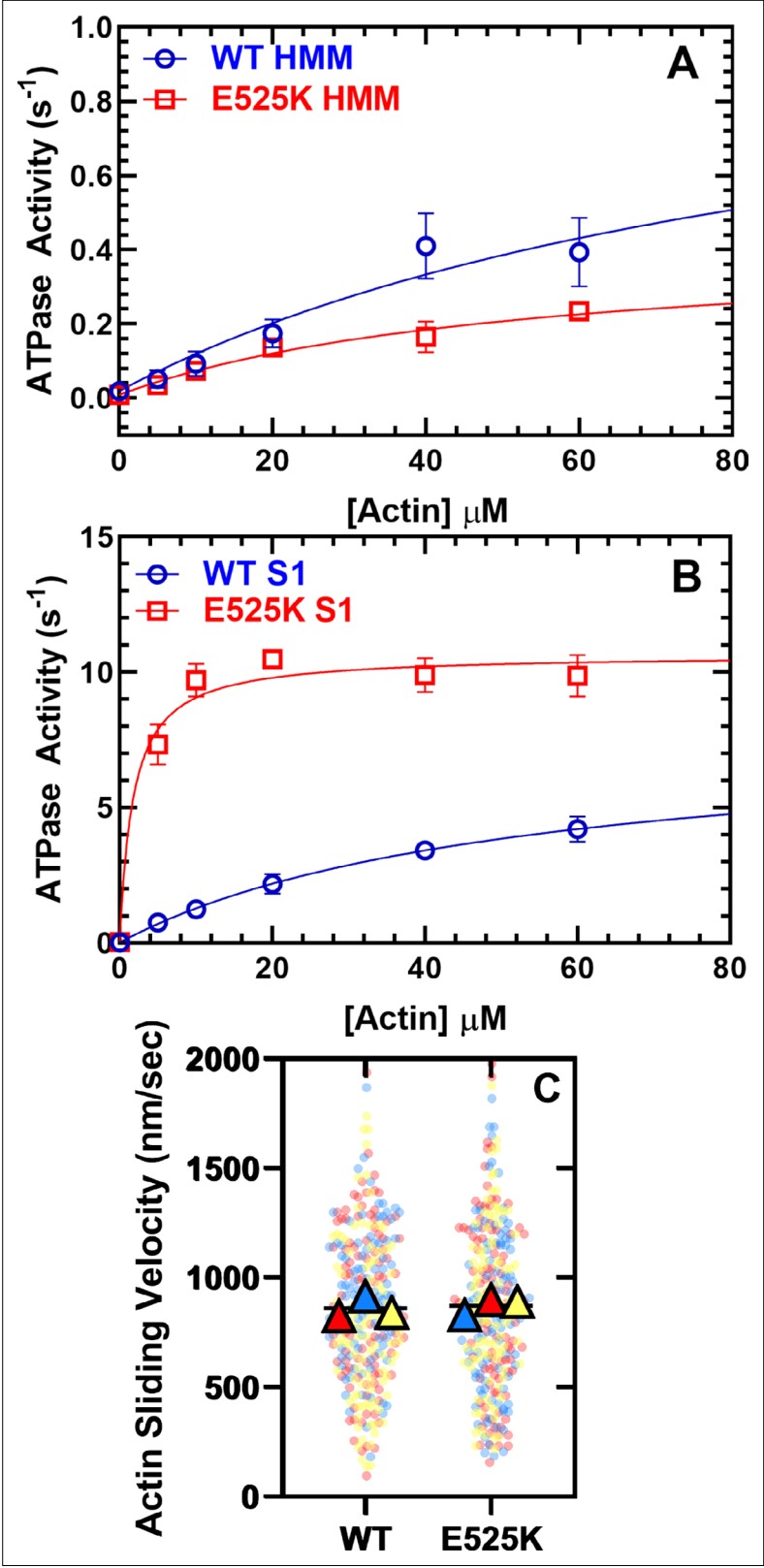

**Figure 2.** Steady-state actin-activated ATPase activity and in vitro motility. (**A**) The ATPase activity of WT and E525K M2β 15HPZ.GFP (heavy meromyosin [HMM]) was measured as a function of actin concentration, and the data were fit to a hyperbolic function to determine $k_{cat}$ and $K_{ATPase}$. (**B**) The ATPase activity of WT and E525K M2β S1 (S1) was also measured as in panel A. (**C**) In vitro motility assays were used to examine actin gliding velocities of HMM

*Figure 2 continued on next page*

*Figure 2 continued*

constructs. The individual data points are plotted for three protein preparations, shown in three different colors (yellow, blue, and red), and the average for each preparation is represented by the colored triangles. The overall average velocity is represented by the black line. Measurements are from at least three protein preparations (± SD). See *Table 1* for summary of values and statistical comparisons.

The online version of this article includes the following video, source data, and figure supplement(s) for figure 2:

**Source data 1.** Excel files with data from *Figure 2*.

**Figure supplement 1.** SDS-gel of M2β S1 and 15HPZ.

**Figure supplement 1—source data 1.** Uncropped SDS-PAGE of WT and E525K S1 and 15HPZ.GFP.

**Figure 2—video 1.** In vitro motility assay.

https://elifesciences.org/articles/77415/figures#fig2video1

---

the SRX state even under physiological ionic strength conditions. The observed rate constants for the fast and slow components of the transients were relatively similar (~within twofold) in WT and E525K M2β 15HPZ.GFP (*Figure 3D* and *Table 2*).

We also monitored the mantATP single turnover rate constants in WT and E525K M2β S1. We found that the slow phase that represents the SRX state was a small fraction of the fluorescence transients (~5%), was similar in WT and E525K, and was independent of ionic strength (*Figure 3*, *Figure 3—figure supplement 3A-C*, *Table 3*). The rate constants of the SRX and DRX states were two- to fourfold faster in E525K compared to WT M2β S1 (*Figure 3*, *Figure 3—figure supplement 3D*, *Table 3*). The rate constants for the DRX state were faster for S1 compared to 15HPZ.GFP in both WT and E525K, while only the mutant also had a faster SRX rate constant in S1 compared to 15HPZ. GFP (*Figure 3—figure supplement 3E–F*). Our results demonstrate a larger increase in DRX rates for S1 constructs compared to 15HPZ.GFP than previously reported by *Rohde et al., 2018* in proteolytically prepared bovine cardiac HMM. The reasons for this difference are unclear, but they are not due to protein quality based on the $NH_4$ ATPase results which demonstrate nearly identical ATPase values for S1 and 15HPZ.GFP HMM.

## Cy3ATP binding to M2β 15HPZ.GFP monitored by FRET

We first tested the FRET biosensor by examining the fluorescence emission spectra of M2β 15HPZ.GFP in the presence or absence of Cy3ATP in low- (20 mM KCl) and high-salt (150 mM KCl) conditions. We observed donor quenching at low salt (5–10%) in the presence of the acceptor that was attenuated at high salt and a similar trend by monitoring acceptor enhancement (*Figure 4—figure supplement 1*). We further characterized the IHM FRET biosensor by mixing Cy3ATP (0.25–2.5 µM) with M2β 15HPZ. GFP (0.25 µM) in a stopped-flow apparatus and monitoring the quenching of the GFP fluorescence as a function of time. The approach allowed us to measure the degree of donor quenching (amplitude) as well as the rate constants for Cy3ATP binding ($k_{obs}$) to M2β 15HPZ.GFP. The FRET transients clearly demonstrated donor quenching in the presence of the acceptor, while donor only controls exhibited

---

**Table 1.** Summary of steady-state ATPase and in vitro motility measurements (± SE).

| Parameter | WT M2β 15HPZ.GFP (N=3–5) | E525K M2β 15HPZ.GFP (N=3) | WT M2β S1 (N=3–6) | E525K M2β S1 (N=3) |
|---|---|---|---|---|
| $v_0$ (s$^{-1}$) | 0.02±0.01, N=5 | 0.01±0.01 | 0.02±0.01, N=6 | 0.03±0.01 |
| $k_{cat}$ (s$^{-1}$) | 1.1±0.49, N=5 | 0.41±0.09 | *7.8±0.7, N=6 | †10.6±0.3 |
| $K_{ATPase}$ (µM) | 99.9±64.5, N=5 | 53.3±20.8 | 52.2±8.6, N=6 | †1.8±0.4 |
| Average sliding velocity (nm/s) | 853±54, N=3 | 852±58 | *,‡1591±16, N=3 | N.D. |
| Percent stuck filaments | 27.8±9.5, N=3 | 34.8±5.4 | ≤5, N=3 | N.D. |

*p<0.005 comparing WT M2β S1 to WT M2β 15HPZ.GFP.
†p<0.05 comparing E525K M2β S1 to WT M2β S1 (unpaired Student's t-test).
‡Data is from *Tang et al., 2021*.

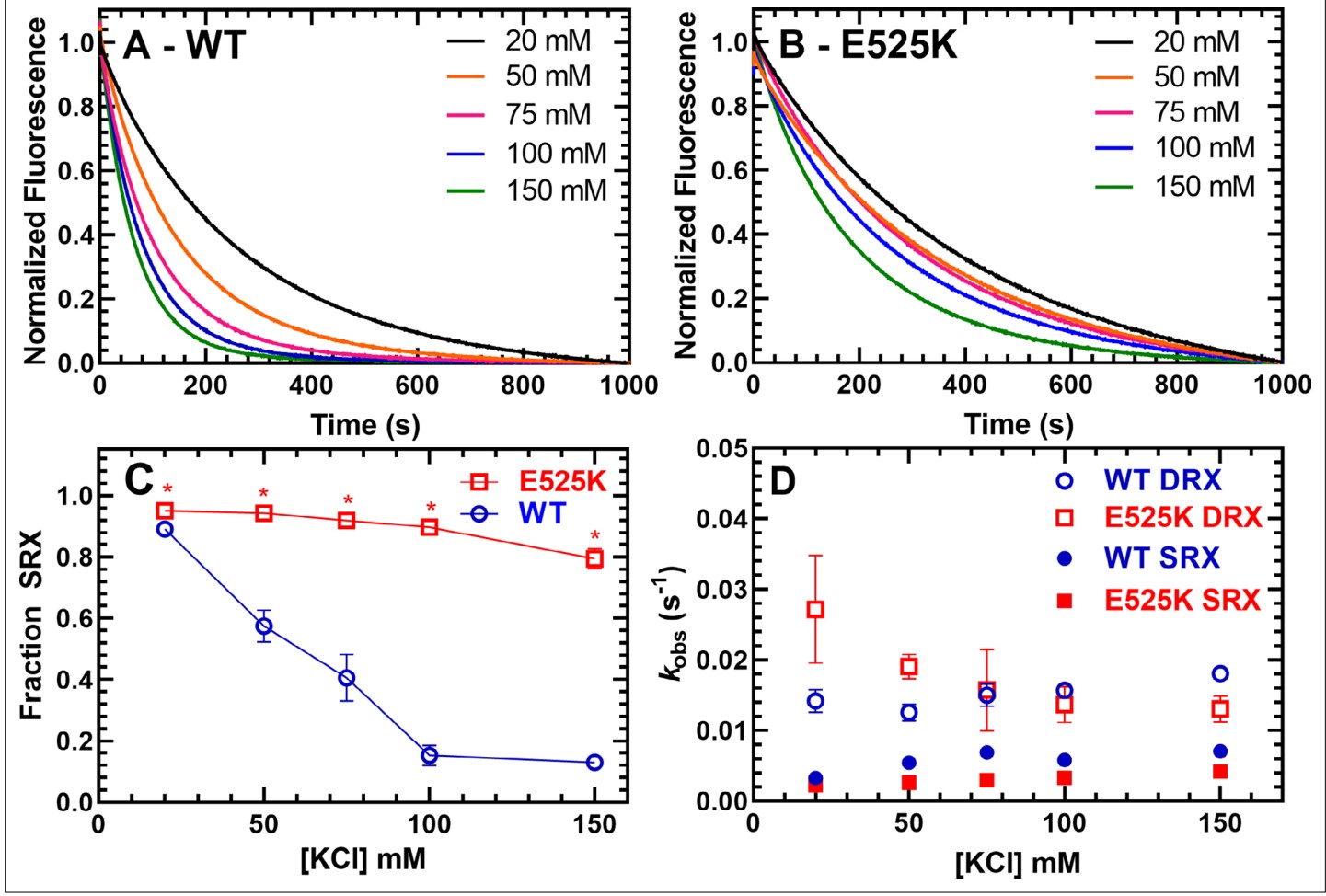

**Figure 3.** Single ATP turnover measurements. The turnover of mantATP by M2β 15HPZ.GFP was examined at varying KCl concentrations. The fluorescence transients from (**A**) WT or (**B**) E525K M2β 15HPZ.GFP (0.25 μM) that was pre-incubated with mantATP (1 μM) for ~30 s and then mixed with saturating unlabeled ATP (2 mM) were observed for 1000 s. The mant fluorescence transients were best fit to a two-exponential function. (**C**) The relative amplitude of the slow rate constants from the fluorescence transients was used to determine the fraction of heads in the super-relaxed (SRX) state. The mutant had a significantly greater SRX state fraction than WT at all KCl concentrations. (**D**) The slow-rate constants (SRX state) were 5–10 times slower than the fast-rate constants (disordered relaxed [DRX] state), and relatively similar (~within twofold) at each KCl concentration for both the WT and E525K constructs. Error bars are ± SD, N=3 separate protein preparations with one technical replicate per prep (*p<0.005, comparing WT and E525K, unpaired Student's t-test). See *Table 2* for summary of values.

The online version of this article includes the following source data and figure supplement(s) for figure 3:

**Source data 1.** Excel files with data from *Figure 3*.

**Figure supplement 1.** Single turnover with WT M2β 15HPZ.

**Figure supplement 1—source data 1.** Excel files with data from *Figure 3—figure supplement 1*.

**Figure supplement 2.** Single turnover with E525K M2β 15HPZ.

**Figure supplement 2—source data 1.** Excel files with data from *Figure 3—figure supplement 2*.

**Figure supplement 3.** Single ATP turnover measurements with M2β S1.

**Figure supplement 3—source data 1.** Excel files with data from *Figure 3—figure supplement 3*.

no change in fluorescence (*Figure 4A&B*). We reasoned that if more myosin heads were in the IHM conformation, we would observe a higher FRET efficiency. The fluorescence transients fit well to a two-exponential function especially at low salt (20–75 mM KCl) with some fitting better to a single exponential function at higher salt (100–150 mM KCl; *Figure 4A&B*). We plotted the total amplitude of the FRET change upon Cy3ATP binding as a function of Cy3ATP concentration and found that WT had a smaller total amplitude at high salt (150 mM) compared to low salt (20 mM; *Figure 4C*). However, the

**Table 2.** Summary of single mantATP turnover measurements for WT and E525K M2β 15HPZ.GFP (N=3,± SD).

| Conditions | Disordered relaxed (DRX) rate | DRX fraction | Super-relaxed (SRX) rate | SRX fraction |
|---|---|---|---|---|
| **WT** | | | | |
| 20 mM KCl | 0.014±0.002 | 0.11±0.01 | 0.0033±0.0001 | 0.89±0.01 |
| 50 mM KCl | 0.013±0.001 | 0.43±0.05 | 0.0054±0.0005 | 0.57±0.05 |
| 75 mM KCl | 0.015±0.002 | 0.59±0.08 | 0.0069±0.0009 | 0.41±0.08 |
| 100 mM KCl | 0.016±0.001 | 0.85±0.03 | 0.0058±0.0004 | 0.15±0.03 |
| 150 mM KCl | 0.018±0.001 | 0.87±0.02 | 0.0071±0.0008 | 0.13±0.02 |
| **E525K** | | | | |
| 20 mM KCl | 0.027±0.008 | 0.05±0.01 | 0.0022±0.0002 | 0.95±0.01 |
| 50 mM KCl | 0.019±0.002 | 0.06±0.01 | 0.0026±0.0001 | 0.94±0.01 |
| 75 mM KCl | 0.016±0.006 | 0.08±0.02 | 0.0030±0.0001 | 0.92±0.02 |
| 100 mM KCl | 0.014±0.003 | 0.10±0.02 | 0.0033±0.0001 | 0.90±0.02 |
| 150 mM KCl | 0.013±0.002 | 0.21±0.03 | 0.0042±0.0001 | 0.79±0.03 |

total amplitudes with the E525K experiments were quite insensitive to KCl concentration (*Figure 4D*), suggesting more M2β 15HPZ.GFP molecules were in the IHM conformation at higher salt. All of the rate constants were linearly dependent on Cy3ATP concentration (*Figure 4E and F*), suggesting they represent second-order rate constants for ATP binding (*Table 4*). The relative amplitude of the fast phase was higher in WT (65%) than E525K (25%) at low salt (*Table 4*, *Figure 4—figure supplement 2*). Overall, the IHM FRET experiments followed a similar trend compared to the single ATP turnover experiments, suggesting the SRX biochemical state correlates with the IHM structural state.

The kinetic results can be interpreted with two possible mechanisms (*Figure 4—figure supplement 3*). In the first mechanism (Scheme 1), HMM can adopt two conformations in the absence of nucleotide (Apo), one that binds ATP with a similar rate constant to S1 (referred to as Open) and another that binds ATP more slowly (referred to as alternate conformation, ALT). Once the Open conformation binds ATP ($K_T$), it rapidly transitions into the IHM ($K_{IHM}$). In this pathway, ATP binding, followed by rapid transition into the IHM, gives rise to the fast phase of the FRET transients. The alternative conformation

**Table 3.** Summary of single mantATP turnover measurements for WT and E525K M2β S1 (N=3, ± SD).

| Conditions | Disordered relaxed (DRX) rate | DRX fraction | Super-relaxed (SRX) rate | SRX fraction |
|---|---|---|---|---|
| **WT** | | | | |
| 20 mM KCl | 0.039±0.002 | 0.95±0.03 | 0.0075±0.0074 | 0.05±0.03 |
| 50 mM KCl | 0.036±0.003 | 0.98±0.01 | 0.0058±0.0045 | 0.02±0.01 |
| 75 mM KCl | 0.034±0.003 | 0.97±0.01 | 0.0082±0.0033 | 0.03±0.01 |
| 100 mM KCl | 0.032±0.003 | 0.97±0.01 | 0.0078±0.0047 | 0.03±0.01 |
| 150 mM KCl | 0.028±0.002 | 0.97±0.01 | 0.0044±0.0030 | 0.03±0.01 |
| **E525K** | | | | |
| 20 mM KCl | 0.085±0.005 | 0.94±0.03 | 0.0197±0.0025 | 0.06±0.03 |
| 50 mM KCl | 0.080±0.005 | 0.95±0.01 | 0.0204±0.0048 | 0.05±0.01 |
| 75 mM KCl | 0.076±0.006 | 0.94±0.04 | 0.0233±0.0012 | 0.06±0.04 |
| 100 mM KCl | 0.071±0.005 | 0.96±0.01 | 0.0185±0.0045 | 0.04±0.01 |
| 150 mM KCl | 0.074±0.018 | 0.97±0.02 | 0.0129±0.0026 | 0.03±0.02 |

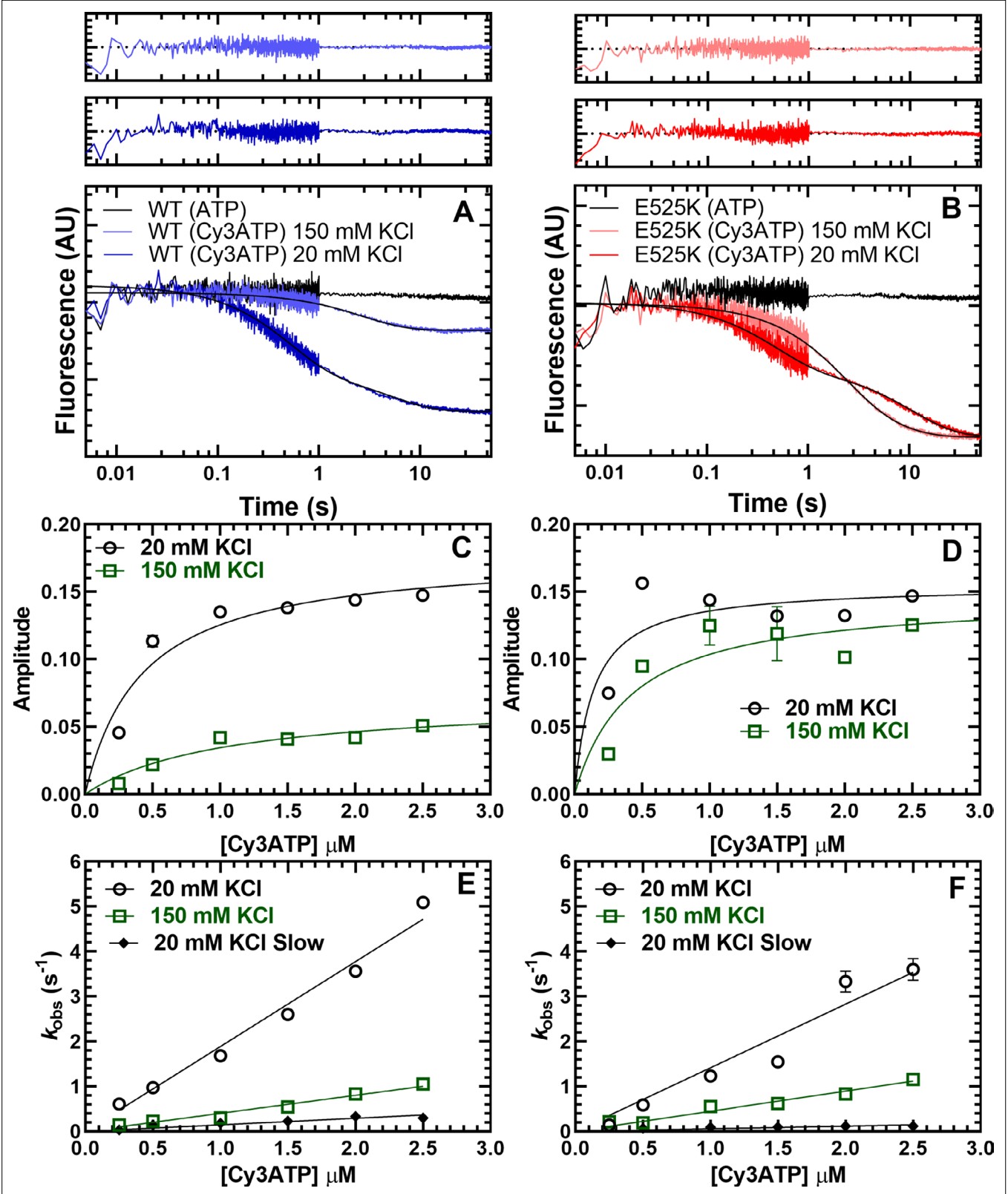

**Figure 4.** Cy3ATP binding to M2β 15HPZ.GFP monitored by FRET. GFP fluorescence (0.25 μM; donor) quenching was examined upon Cy3ATP (1 μM; acceptor) binding to WT (**A**) or E525K (**B**) M2β 15HPZ.GFP. The fluorescence transients were best fit to a double-exponential function at low salt (20 mM KCl) and a single or double exponential function at high salt (150 mM KCl; fast phase was 65% and 25% of the signal at low salt in WT and E525K, respectively). Residuals are shown above each plot. The total amplitude of the fluorescence change was plotted as a function of Cy3ATP concentration

*Figure 4 continued on next page*

*Figure 4 continued*

in WT (**C**) and E525K (**D**). All rate constants were linearly dependent on Cy3ATP concentration in both WT (**E**) and E525K (**F**) M2β 15HPZ.GFP (see *Table 4* for summary of linear fits in panel E and F). Each data point in C–F represents 2–3 replicates from a single preparation of myosin, and errors bars indicate ± SE of the fit.

The online version of this article includes the following source data and figure supplement(s) for figure 4:

**Source data 1.** Excel files with data from *Figure 4*.

**Figure supplement 1.** Steady-state FRET spectra.

**Figure supplement 1—source data 1.** Excel files with data from *Figure 4—figure supplement 1*.

**Figure supplement 2.** Amplitudes for Cy3ATP binding to M2β 15HPZ.

**Figure supplement 2—source data 1.** Excel files with data from *Figure 4—figure supplement 2*.

**Figure supplement 3.** Kinetic models of interacting-heads motif (IHM) formation based on stopped-flow FRET.

**Figure supplement 3—source data 1.** Excel files with data from *Figure 4—figure supplement 3*.

**Figure supplement 4.** FRET transients fit to the proposed kinetic model.

**Figure supplement 4—source data 1.** Excel files with data from *Figure 4—figure supplement 4*.

binds ATP about 10–20-fold slower ($K'_T$) and then can rapidly transition into the IHM ($K'_{IHM}$), thus giving rise to the slow phase that is also ATP concentration dependent. In this model, FRET is observed only in the IHM. Representative fluorescence transients from experiments with WT and E525K at low- and high-salt fit reasonably well to this kinetic model (*Figure 4—figure supplement 4*). Another possible mechanism is that HMM can undergo a slow isomerization between two conformations in the Apo state, an alternative conformation incompetent to bind nucleotide (INC) and an Open conformation (Scheme 2). In this model, the slow phase is caused by the slow isomerization into the Open state prior to ATP binding. This mechanism was originally proposed by *Geeves et al., 2000* in a study on Myo1C and was since demonstrated in several other studies (*Adamek et al., 2010*; *Clark et al., 2005*; *Ušaj and Henn, 2017*). Both models predict the E525K mutant stabilizes an alternate or incompetent Apo conformation at low salt, which explains the higher percentage of the slow phase observed in E525K. In addition, both models predict that high-salt favors the open Apo conformation in WT and mutant, while the transition into the IHM is unfavorable for WT but favorable for E525K.

## Steady-state and time-resolved FRET measurements

The stopped-flow fluorescence transients were also used to determine the FRET efficiency after performing donor only controls. Measuring a precise distance by FRET can be quite challenging and is most accurate when the distance is between 0.5 and 1.5 $R_0$ (31.5–94.5 Å). Since we predicted our closest distance in the IHM is ~100 Å (*Figure 1*), we emphasize our FRET approach is best utilized for measuring the change in FRET associated with different KCl concentrations and comparing WT and E525K, while it is not appropriate for measuring precise distances. We measured the FRET efficiency (0.25 μM M2β and 1 μM Cy3ATP) as a function of KCl concentration and found that WT shifts to a lower FRET state at higher KCl concentrations, while E525K maintains a similar FRET efficiency at low and high salt (*Figure 5A*). We then compared the measured FRET efficiencies to the expected efficiencies assuming a simplified model in which the fraction of molecules in the closed and open states

**Table 4.** Summary of kinetic parameters from Cy3ATP binding experiments. (N=1,± SE of the fit).

| Parameter | WT | E525K |
|---|---|---|
| $k_{fast}$ (20 mM KCl) (μM$^{-1}$ s$^{-1}$) | 1.9±0.1 | 1.4±0.1 |
| $k_{slow}$ (20 mM KCl) (μM$^{-1}$ s$^{-1}$) | 0.15±0.01 | 0.06±0.01 |
| $A_{fast}$ (20 mM KCl) | 0.65±0.04 | 0.25±0.05 |
| $k_{fast}$ (150 mM KCl) (μM$^{-1}$ s$^{-1}$) | 0.40±0.02 | 0.45±0.02 |
| $k_{slow}$ (150 mM KCl) (s$^{-1}$) | 0.06±0.03 | 0.12±0.06 |
| $A_{fast}$ (150 mM KCl) | 0.81±0.03 | 0.89±0.16 |

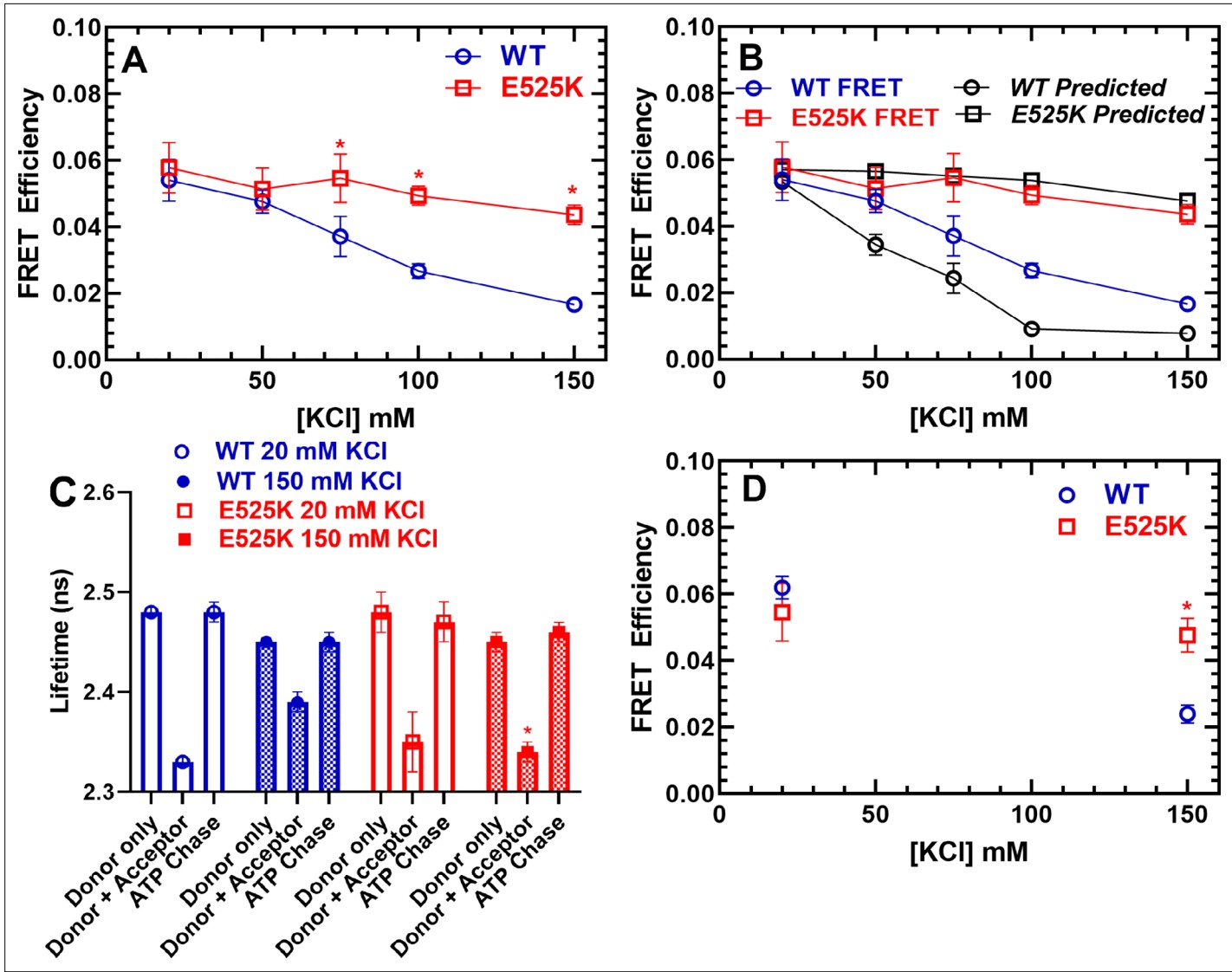

**Figure 5.** FRET monitors stability of the closed (interacting-heads motif [IHM]) conformation. Steady-state FRET between the C-terminal GFP tag and Cy3ATP bound to the motor domain was examined in the stopped-flow as a function of KCl concentration. (**A**) The average FRET efficiency (± SD) was examined in three separate protein preparations. (**B**) The FRET efficiency in panel A was compared to the predicted FRET efficiency, based on the single turnover measurements in *Figure 3*. Error bars in panels A and B are ± SD, N=3 separate protein preparations with three technical replicates per prep. (**C**) Time-resolved FRET was examined by monitoring the GFP fluorescence lifetime in the presence and absence of Cy3ATP. Summary of fluorescence lifetime data with WT and E525K M2β 15HPZ.GFP. Adding saturating ATP to the sample containing WT M2β 15HPZ.GFP and Cy3ATP (ATP chase) was used to rule out donor fluorescence changes in the presence of ATP. (**D**) Time-resolved FRET (three technical replicates performed with a single protein preparation) demonstrated similar results to steady-state FRET shown in panel A (*p<0.05 comparing WT and E525K, unpaired Student's t-test).

The online version of this article includes the following source data and figure supplement(s) for figure 5:

**Source data 1.** Excel files with data from *Figure 5*.

**Figure supplement 1.** Representative fluorescence lifetime decays for TR-FRET analysis.

**Figure supplement 1—source data 1.** Excel files with data from *Figure 5—figure supplement 1*.

directly correlates with the single turnover measurements (*Figure 5B*). We assumed no FRET in the open state and a FRET efficiency of ~6.0% associated with a 100 Å distance when all heads are in the IHM configuration. The expected FRET efficiencies followed a similar trend to the measured efficiencies, especially in the E525K mutant, while the largest discrepancies were observed with WT in higher salt conditions. This is likely due to the greater degree of uncertainty with the measurements at high salt, which contained the lowest FRET efficiencies.

We also performed time-resolved FRET to further verify the differences we observed between WT and E525K (*Figure 5C and D*). Time-resolved FRET relies on monitoring the donor fluorescence lifetime decay in the presence and absence of acceptor and thus is less sensitive to sample-to-sample variability in fluorophore concentration. We examined the GFP fluorescence lifetime (60 nM M2β 15HPZ.GFP) in the absence of nucleotide and then again after adding 0.1 µM Cy3ATP (*Figure 5—figure supplement 1*). We then added saturating unlabeled ATP (1 mM) to chase off the Cy3ATP as a control for changes in donor fluorescence in the presence of ATP (*Figure 5C*). The fluorescence decays were fit well to a two-exponential function which was used to determine the average lifetime in each condition (*Figure 5—figure supplement 1*). The donor fluorescence lifetimes were unaffected by the presence of ATP (*Figure 5C*). Overall, we found that the time-resolved and steady-state FRET measurements resulted in very similar FRET efficiencies (*Figure 5A and D*).

## Negative stain EM

WT and E525K M2β 15HPZ.GFP constructs were examined by negative stain EM to directly observe the impact of the mutation on the configuration of the myosin heads and to test our interpretation of the FRET data. Because the IHM is easily disrupted by binding to the grid surface during specimen preparation (*Burgess et al., 2007*), we examined the molecules in low-salt (20 mM KCl) conditions, which strengthen the IHM, and minimized the time of contact of protein with the grid before staining. We performed the analysis on three separate preparations. The comparison of WT and E525K for each individual replicate was performed on the same day using grids from the same batch, prepared identically (glow-discharge etc.) under the same ambient conditions of temperature and humidity. WT molecules were dominated by non-interacting heads, producing an 'open' conformation (*Figure 6A*, yellow circles; *Figure 6D*), with only a small number of 'closed' (folded) IHM structures (*Figure 6A*, green circles; *Figure 6D*). In contrast, more of the E525K molecules adopted the folded (IHM) conformation (*Figure 6B*, green circles; *Figure 6D*). Each replicate was performed on a different day, therefore, with non-identical conditions between preps. This likely accounts for differences in numbers of molecules in open and closed configurations between preps, as we know that conformation is highly sensitive to grid surface properties. Despite the experimental 'noise' introduced by these factors, there was a clear and dramatic 3-4 fold increase in the number of closed conformations in E525K compared with WT, consistent with stabilization of the IHM by the mutation and with the FRET data (*Figure 6D*, *Table 5*). Because of the strong influence of grid surface properties on conformation, disrupting the weak interactions of the IHM, a quantitative agreement with FRET data is not expected. For each prep of WT and E525K, a total of ~250–300 molecules were counted. Molecules that could be classified as open or closed were 70–85% of the total (*Table 5*); the remainder did not show a clearly identifiable structure (white circles in *Figure 6A and B*).

To validate our assessment of open and closed percentages based on visual inspection of the raw micrographs (*Figure 6A, B and D*), we performed 2D class averaging of the molecules using Relion 3.1 (*Scheres, 2012*; *Figure 6—figure supplement 1*). The class averages clearly show that in the case of E525K, more of the molecules form the IHM conformation when compared with WT (*Figure 6—figure supplement 1A and B*). We also used class averaging to confirm that the closed structures we observed were actually IHMs. The E525K class average shown in *Figure 6C* reveals key features of the IHM, including the BHs and FHs, the two light chains (essential light chains, ELCs, and regulatory light chains, RLCs), and sub-fragment 2 interacting with the BH, all well-documented features of the IHM (*Figure 6—figure supplement 1C–E*). We conclude that the raw EM images and the 2D class averages qualitatively support our ATP turnover and FRET results showing that E525K promotes the IHM and thus stabilizes the SRX state.

## Discussion

A common mechanism of myosin II regulation is via folding of the heads back onto the tail and interaction between the motor domains, effectively inhibiting actomyosin interactions and myosin ATP turnover. In the current study, we investigated the regulatory mechanism in cardiac muscle myosin, which exists in dynamic equilibrium between the folded-back conformation (IHM) and an open conformation capable of generating force upon muscle activation (*Figure 1*). A major question in the field is whether the IHM correlates with the SRX state, a biochemical state that turns

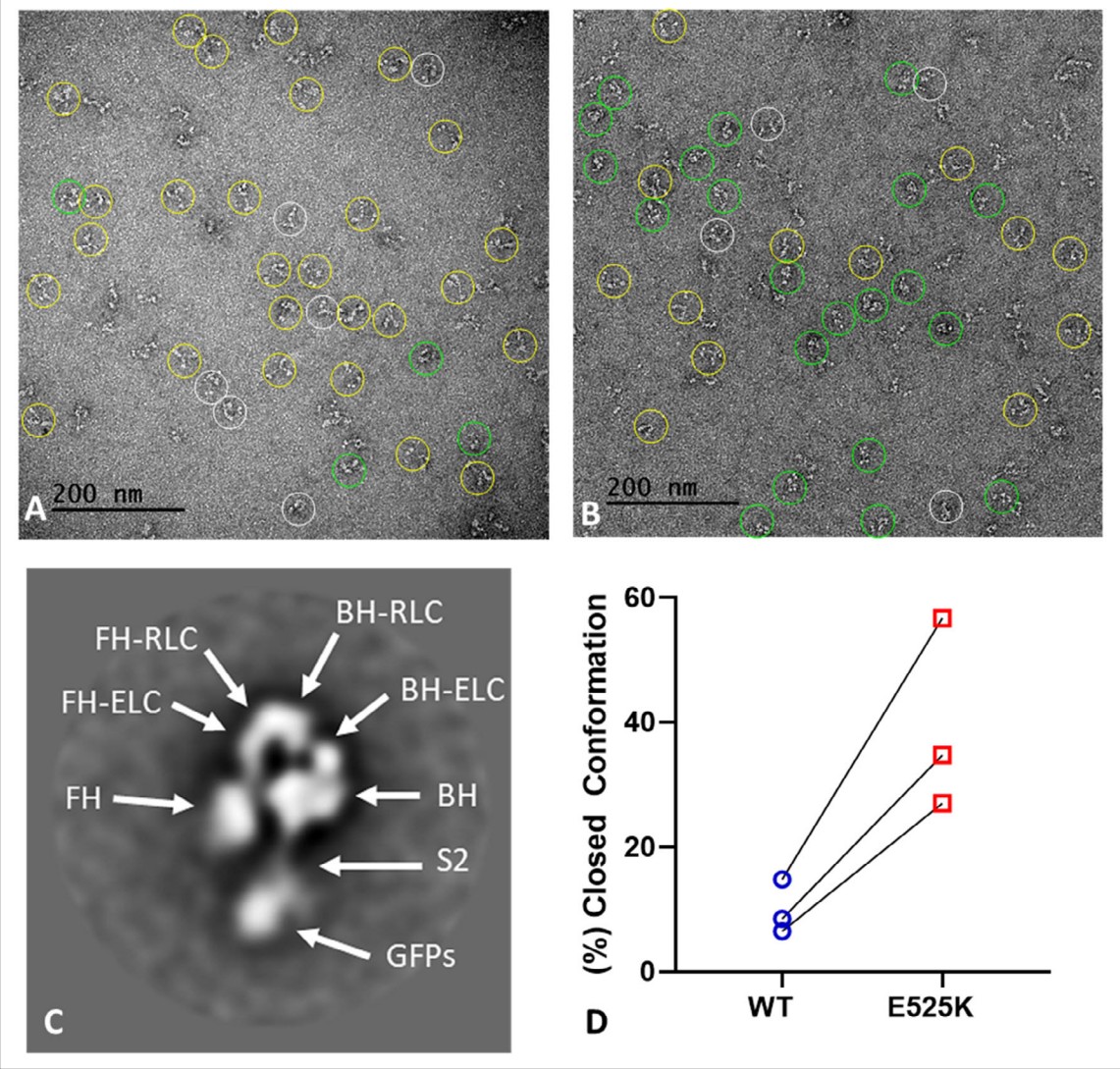

**Figure 6.** EM structure of WT and E525K constructs. (**A** and **B**) Negative stain images of WT and E525K, respectively, showing the conformational state of the molecules (yellow – open; green – closed; white circle – not clear/partially folded). (**C**) 2D class average of E525K, showing subdomains (*Figure 1*, 'closed'). (**D**) Results of three biological replicate experiments comparing % of closed (interacting-heads motif [IHM]) conformations in WT and E525K. We attribute the variable results to unavoidable differences in grid surface characteristics on the different experimental days. Despite these differences, the same trend occurred in each experiment, with E525K showing 3–4 times as many IHMs as WT (see *Table 5* for summary, N=250-300 molecules per preparation).

The online version of this article includes the following source data and figure supplement(s) for figure 6:

**Source data 1.** Electron micrographs used for classifying molecules in the folded-back interacting-heads motif (IHM) or open conformation .

**Figure supplement 1.** 2D class averaging of M2β 15HPZ.

**Figure supplement 2.** Potential intramolecular interactions of E525K with S2.

**Table 5.** Summary of negative stain EM images classified as open vs. closed conformation.

| Preparation | WT (% closed) | WT (% open) | E525K (% closed) | E525K (% open) | % E525K closed/ % WT closed |
|---|---|---|---|---|---|
| 1 | 16.9 | 83.1 | 56.9 | 43.1 | 3.4 |
| 2 | 8.5 | 91.5 | 34.8 | 65.2 | 4.1 |
| 3 | 6.5 | 93.5 | 27.0 | 73.0 | 4.2 |

over ATP 5–10-fold slower than single-headed S1. This proposed mechanism is attractive since the sequestered heads would inhibit interactions with actin, serving as a reserve to be recruited when enhanced contractile force is needed (e.g. length-dependent activation). Furthermore, sequestered heads would consume far less ATP, helping to balance the energetic demands of the heart. We present direct evidence that the fraction of myosin heads in the IHM correlates well with the fraction of myosin heads in the SRX state, providing crucial information about the structural basis of the auto-inhibited state of cardiac myosin. We also demonstrate that a dimeric cardiac myosin molecule with 15 heptads of the coiled-coil displays the SRX state and IHM structure, allowing us to examine the folded conformation with a novel IHM FRET biosensor. In addition, we report that a single point mutation associated with DCM (E525K) can dramatically stabilize both the IHM conformation and SRX state. Our work highlights the stabilization of the IHM by electrostatic interactions and the critical function of the E525 residue in the core region where electrostatic interactions between the head and tail occur. The clinical significance of this study is that a DCM mutation can stabilize the auto-inhibited state of myosin, which provides a strong rationale for studying other DCM mutants to determine if this is a common mechanism that causes the reduced muscle force and power typically observed in DCM patients.

## Correlating the SRX and IHM

The current study provides new information about the structural basis of the slow ATP turnover (SRX) state. Previously, *Anderson et al., 2018* demonstrated a strong correlation between the IHM and SRX state using negative stain EM and single ATP turnover measurements. They also examined the drug mavacamten, which dramatically stabilizes the SRX state. Based on visual inspection of negative stained EM images, they observed a corresponding increase in the fraction of molecules that adopt the IHM in the presence of the drug. Our study used single turnover measurements, in addition to an IHM FRET biosensor, to examine the conformation of cardiac myosin in solution. The fraction of myosin heads in the SRX state determined by single turnover measurements was used to simulate the FRET results expected if the SRX state and IHM are 1–1 correlated (*Figure 5B*). We generated FRET efficiency estimates based on the structural model shown in *Figure 1*, which predicts the average distance between donor-acceptor pairs in the IHM is in the range of 90–140 Å and no FRET in the open state. The measured FRET efficiency (5.5%) observed in low-salt conditions (*Figure 5A*) corresponded to a condition with nearly 90% SRX heads (*Figure 3C*, *Table 2*), which allowed us to assume an efficiency of 6.0% in the IHM for our simulations. The simulated data follows fairly closely with the measured FRET efficiencies for the mutant but diverges for WT at higher salt (≥100 mM KCl; *Figure 5B*), likely because the fraction of myosin heads in the IHM is quite low (≤10%) in high-salt conditions and difficult to measure by FRET. Future experiments using time-resolved FRET approaches such as TR(2)-FRET (*Gunther et al., 2019*; *Rohde et al., 2018*) may allow us to determine distance distributions, distances associated with partially folded intermediates, and the rate constants for transitioning into and out of the IHM.

Another study performed single turnover measurements with a different IHM FRET biosensor in bovine cardiac myosin and found the IHM was sufficient but not required to produce the SRX state (*Chu et al., 2021*). They used glutaraldehyde cross linking to stabilize the IHM and found a corresponding increase in FRET. They demonstrated that mavacamten dramatically stabilizes the SRX state but did not cause a corresponding increase in the IHM formation, suggesting that the biochemical SRX state and structural IHM are uncoupled. Importantly, their FRET approach was not sensitive to ionic strength, which is very different from our results and other previous IHM studies. Other groups also have demonstrated that myosin constructs that lack an S2 domain, such as S1 and 2HP HMM (and thus cannot form the IHM), still have a small SRX state component in single turnover measurements (*Anderson et al., 2018*; *Rohde et al., 2018*). Our results fit well with the hypothesis that the SRX state may not require the formation of the IHM, but the IHM certainly enhances the SRX state by stabilizing the SRX heads (*Craig and Padrón, 2022*). The fraction of SRX heads also depends on their supramolecular organization, with the SRX molecules increasing from M2β S1 to HMM to a thick filament (*Craig and Padrón, 2022*). This relationship accounts for nominal SRX ratios observed in S1 and 2HP HMM as well as the correlation between SRX and IHM in our M2β HMM 15HPZ.GFP biosensor. Moreover, it accounts for the high sensitivity of the SRX state to ionic strength in HMM since many of the proposed interactions in the IHM are electrostatic in nature.

In addition to using FRET to monitor IHM formation, we examined negative stain EM images of M2β HMM 15HPZ.GFP, both by direct visual inspection and 2D class averaging, which allowed us to systematically classify the fraction of molecules in the IHM. We found that the E525K mutant had a much higher fraction of myosin heads in the IHM compared to WT (*Figure 6*). However, this fraction was significantly different in the EM images compared to the predictions from the FRET and single turnover measurements in solution. We believe this is due to molecular interactions with the EM grid surface that destabilize the IHM. This has been well documented with smooth muscle HMM (*Burgess et al., 2007*). Our similar but even more extreme finding with cardiac myosin is exactly as predicted, given the lower stability of the cardiac compared with the smooth muscle myosin IHM (*Jung et al., 2008*). In fact, we found it essential to reduce the time of contact of the cardiac construct with the grid surface to a minimum (~5 s) before adding the uranyl acetate stain, which then immediately fixes the structure (*Zhao and Craig, 2003*). With longer contact times, almost all IHMs were lost, consistent with the smooth muscle HMM findings (*Burgess et al., 2007*). Indeed, a recent study that performed small angle x-ray scattering on a cardiac myosin 25 heptad construct found that alternative conformations are likely in solution, including partially folded intermediates (e.g. one head folded and one head free; *Gollapudi et al., 2021*). We observed many structures by 2D class averaging with different orientations of heads. However, it is difficult to distinguish between novel conformations and states that simply lie on the grid in different orientations. Overall, our work clearly demonstrates by both EM and FRET that the IHM is a major contributor to the SRX state, while our EM studies suggest that the IHM is quite dynamic and sensitive to surface interactions.

## Kinetics of IHM formation

One unanswered question is how fast various myosins can transition into and out of the IHM. Geeves and co-workers estimated the rate of transition into the IHM was at least 5 s$^{-1}$ in bovine cardiac myofibrils (*Walklate et al., 2022*). In agreement, our results suggest the transition into the IHM was significantly faster than ATP binding (≥5 s$^{-1}$; *Figure 4*). Our results also suggest there are two conformations of HMM in the absence of nucleotide, one that binds ATP similar to S1 (Open) and one that binds ATP about 10–20-fold slower (ALT; see *Figure 4—figure supplement 3*). We also found that high salt shifts the equilibrium toward the Open conformation in the absence of nucleotide for both WT and E525K, while only the mutant can transition efficiently into the IHM upon ATP binding. Alternatively, the data may suggest that there is an isomerization between competent and incompetent states in the absence of nucleotide, which has been proposed to explain the bi-exponential kinetics of ATP binding to myosin in previous studies (*Geeves et al., 2000*; see *Figure 4—figure supplement 3*). Thus, further work will be necessary to define the specific structural and kinetic pathway for transition into and out of the IHM, and current and future IHM FRET biosensors could lead the way in this important area of study.

## Structural mechanism of IHM and impact of E525K

Presently, there is no high-resolution structure of the cardiac myosin IHM. The work of Padrón and colleagues has created a model based on the tarantula thick filaments (5TBY) but using the human cardiac myosin sequence, which has been extremely useful for mapping potential head-head and head-tail interactions (*Alamo et al., 2017*). The recent cryo-EM structures of smooth muscle myosin in the 10 S conformation provide more models of the IHM, which have some similarities and some differences compared to 5TBY (*Heissler et al., 2021*; *Scarff et al., 2020*; *Yang et al., 2020*).

The E525K mutation was identified in a genetic study as a de novo variant of high clinical significance (*Lakdawala et al., 2012a*). Previous structural analysis of cardiac myosin further implicates the mutation due to its location on a flat, broad region of myosin known as the mesa (*Spudich, 2015*). The E525 residue is located in the lower 50 kDa region of the myosin motor domain in Helix Q of the helix-loop-helix (*Colegrave and Peckham, 2014*), which is known to be crucial for actin binding (*Várkuti et al., 2012*). From the perspective of the myosin mesa, E525 is in the central region of the 'mesa trail,' a region of the mesa shown to interact directly with S2 in formation of the IHM (*Nag et al., 2017*; *Woodhead and Craig, 2020*). Electrostatic attraction between the mesa trail of the BH and S2 is believed to underlie a crucial BH/S2 priming interaction in the formation of the IHM (*Alamo et al., 2017*). Previous structural analyses indicate that the central mesa trail (rich in positively charged residues) interacts with the Ring 1 region of S2 (rich in negatively charged residues),

and these complementary charged patches likely encourage formation of the IHM. Disruption of the charge distribution in the BH/S2 interaction region could cause complete elimination of the IHM, promoting myosin heads to populate the open or force-generating state, as has been proposed with many HCM mutations (*Sarkar et al., 2020*). E525 is a negatively charged residue within the positively charged mesa trail and may function normally to repel Ring 1/mesa interactions, helping maintain the delicate balance of attractive/repulsive forces required for the precise functioning of the IHM. A charge-reversal point mutation, such as E525K, that increases the positive surface charge of the mesa trail likely would strengthen the Ring 1/mesa interaction in the BH, thus promoting the formation of the IHM. Importantly, E525, as well as many other charged residues in the mesa trail and Ring 1 of S2, is well conserved across species and myosin isoforms, suggesting a critical role for electrostatic interactions between BH/S2 in the formation of the auto-inhibitory state (*Woodhead and Craig, 2020*). Without a high-resolution structure of the cardiac muscle myosin IHM, specific side chain residues in the BH/S2 interaction have not been confirmed. However, our collective data presented here coupled with previous conservation analyses indicate that E525 may be a critical residue in forming the IHM. This is further supported by our molecular modeling of the IHM, in which the introduced lysine residue at 525 of the BH is within several Angstroms of several acidic residues in the S2 region (*Figure 6— figure supplement 2*).

An alternative mechanism by which a mutation can enhance the formation of the IHM is by favoring a conformation of an intermediate that precedes IHM formation (e.g. allosteric mechanism). There is little information about the allosteric mechanisms that mediate IHM formation, while several studies have proposed that the pre-power stroke conformation of the motor domain may favor IHM formation (*Alamo et al., 2017*; *Robert-Paganin et al., 2018*). Our single turnover measurements with E525K and WT S1 suggest that they have a similar ability to populate the SRX state in S1 (~5%), and this small fraction of SRX formation is independent of ionic strength (*Figure 3—figure supplement 3*, *Table 3*). Therefore, we conclude that E525K enhances the IHM and SRX state via electrostatic interactions.

## Implications for understanding the underlying mechanisms of DCM

Determining mutation-associated alterations in cardiac muscle myosin structure and function can lead to novel therapeutic strategies for the treatment of inherited cardiomyopathies such as HCM and DCM. A leading hypothesis is that HCM mutations enhance myosin-associated force and power, while DCM mutations essentially do the opposite (*Debold et al., 2007*; *Spudich, 2014*). So far, mutations in myosin associated with HCM and DCM have been found to have diverse effects on myosin motor function, including altering ATPase activity, actin sliding velocity, duty ratio, and load dependence (*Trivedi et al., 2020*; *Ujfalusi et al., 2018*). However, HCM mutations have been shown to cluster preferentially in the head-head and head-tail interaction interfaces, likely weakening interactions that underlie the SRX state, leading to greater availability of heads for actin interaction and thus to hypercontractility (*Trivedi et al., 2018*). The current study demonstrates that a DCM mutation can do just the opposite, dramatically stabilizing the SRX state, suggesting the possibility that other mutations may have a similar effect. A decrease in the number of available myosin heads in the thick filament would have a profound impact on contractile force and fit well with the hypocontractility hypothesis. Thus, therapeutic strategies that can destabilize the IHM and SRX state would be a logical strategy for treating DCM. However, unlike HCM, which primarily impacts sarcomeric genes, DCM is more heterogeneous and associated with a variety of genes (*Yotti et al., 2019*). Nevertheless, treating DCM with an SRX state destabilizer may be feasible regardless of the underlying genetic cause since it could accomplish the overall goal of enhancing contractility. The E525K mutant provides great insight into an energy conservation mechanism that may underlie the development of the DCM phenotype in this patient population. Moreover, our data indicate that it can serve as a model mutation to deepen our understanding of the SRX biochemical state and IHM structural state in beta-cardiac myosin.

The E525K mutation clearly stabilizes the SRX state and IHM in the M2β 15HPZ.GFP construct, while it enhances the actin-activated ATPase activity in S1 (*Figure 2A and B*). Remarkably, the mutation's ability to stabilize the IHM through electrostatic interactions between the BH and S2 seems to dominate the effects on ATPase activity in M2β 15HPZ.GFP. Future studies with M2β E525K S1 will investigate how this mutation structurally and biochemically enhances intrinsic actin-activated ATPase activity and impacts motile function. Other future investigations will examine how this intriguing

mutation alters ensemble force production, given the enhanced intrinsic ATPase activity of the motor domain but dramatically stabilized SRX/IHM state in M2β 15HPZ.GFP.

It is important to point out some of the limitations and alternative interpretations of the results in the current study. The M2β 15HPZ.GFP was examined with mouse light chains intrinsic to the C2C12 myocyte expression system, which could alter the impact of the E525K mutation. It has been suggested that full-length myosin in a thick filament environment can further stabilize the SRX state through interactions of IHMs with each other along the helical tracks of heads and with other thick filament proteins (*Craig and Padrón, 2022*). Interestingly, mutations measured in purified HMM qualitatively had the same impact on the SRX state in a thick filament preparation (*Gollapudi et al., 2021*). The FRET approach is a biosensor of the IHM structural state, while the open state is not visible by FRET, preventing us from directly determining the mole fraction of the IHM and open states. The FRET distances in the current approach are not in the ideal range ($0.5-1.5$ $R_0$) for measuring precise distances by FRET. Finally, EM provides a snapshot of a fraction of myosin molecules, while the solution measurements (single turnover and FRET) provide information on the entire ensemble. The EM studies also have the limitation that the charge of the EM grid can influence the stability of the IHM.

## Summary

In summary, we examined a human cardiac myosin construct with 15 heptads in its coiled coil and used EM imaging to show that it can form the conserved, auto-inhibited conformation known as the IHM. We also used a novel IHM FRET sensor to monitor the formation of the IHM in solution and determined that it correlates well with the SRX state with slow ATP turnover. Furthermore, we discovered that a DCM charge reversal mutation (E525K) in the motor-tail interface can greatly enhance the stability of the IHM conformation and SRX state. We conclude that this elegant method of regulation, which essentially sequesters myosin heads to balance the mechanical and energetic demands of the heart, is highly sensitive to modifications that alter the electrostatic interactions important for stabilizing the auto-inhibited conformation. Thus, future studies will investigate other DCM mutations that may also stabilize the IHM and SRX state as well as determine how other physiological factors and disease states alter this important regulatory mechanism.

# Materials and methods

## Reagents

Cy3ATP (1 mM stock) and 2'-deoxy-ATP labeled with N-Methylanthraniloyl at the 3'-ribose position (mantATP; 10 mM stock) were purchased from Jena Biosciences. ATP and ADP were prepared from powder (MilliporeSigma), and concentrations were determined by absorbance at 259 nm ($\varepsilon_{259}=15{,}400$ $M^{-1}$ $cm^{-1}$). All assays were performed in MOPS 20 buffer (10 mM MOPS, pH 7.0, 20 mM KCl, 1 mM $MgCl_2$, 1 mM EGTA, and 1 mM DTT) at 25°C unless otherwise noted.

## Protein expression and purification

The human beta-cardiac myosin 15 heptad HMM construct includes residues 1–946 of the *MYH7* gene (GenBank: AAA51837.1) with a leucine zipper GCN4 sequence (MKQLEDKVEELLSKNYHLEN EVARLKKLVGER) added after residue 946, followed by a short linker (GSGKL), a C-terminal EGFP tag (M2β 15HPZ.GFP), Avi (GLNDIFEAQKIEWHE), and FLAG (DYKDDDDK) tags. Another construct was generated of just the S1 region which included residues 1–842 of the *MYH7* gene and C-terminal Avi and FLAG tags (*Swenson et al., 2017*). The E525K mutation was introduced by Quikchange sited directed mutagenesis into M2β 15HPZ.GFP and M2β S1. The constructs were cloned into the pDual shuttle vector, and the initial recombinant adenovirus stock was produced by Vector Biolabs (Malvern, PA, USA) at a titer of $10^8$ plaque forming units per mL (pfu $mL^{-1}$). As previously described, the virus was expanded by infection of Ad293 cells at a multiplicity of infection of 3–5 (*Swenson et al., 2017*). The virus was harvested from the cells and purified by CsCl density sedimentation, giving a final virus titer of $10^{10}-10^{11}$ pfu $mL^{-1}$.

The mouse skeletal muscle derived C2C12 cell line was used to express the cardiac myosin constructs as described previously (*Chow et al., 2002*; *Swenson et al., 2017*; *Wang et al., 2003*; *Winkelmann et al., 2015*). Briefly, C2C12 cells were grown to ~90% confluency on tissue culture plates (145/20 mm) in growth media (DMEM with 9% FBS). On the day of infection, 20 plates were

differentiated by changing the media to contain horse serum (DMEM with 9% horse serum, 1% FBS) and simultaneously infected with virus at $4 \times 10^7$ pfu mL$^{-1}$. The cells were harvested for myosin purification 10–12 days after infection, and M2β constructs were purified by FLAG affinity chromatography. Actin was purified using acetone powder from rabbit skeletal muscle (*Pardee and Spudich, 1982*).

### Steady-state ATPase assays
The actin-activated ATPase of 0.1 µM M2β in the presence and absence of actin (5, 10, 20, 40, and 60 µM) was examined in MOPS 20 buffer at 25°C with the NADH coupled assay as described in previous studies (*Rasicci et al., 2021*; *Swenson et al., 2017*; *Tang et al., 2021*). The ATPase activity was plot as a function of actin concentration and fit to a Michaelis Menten equation to determine the maximum ATPase rate ($k_{cat}$) and actin concentration at which ATPase in one-half maximal ($K_{ATPase}$). At least three preparations (one technical replicate per prep) of each construct were examined.

### NH$_4^+$ ATPase assays
A high-salt ATPase assay was used to evaluate the number of active myosin heads in our preparations (M2βS1 and 15HPZ.GFP). The assay was performed at room temperature (22 ± 1°C) in NH$_4^+$ ATPase buffer (25 mM Tris. 0.4 M NH$_4$Cl, 2 mM EDTA, and 0.2 M sucrose at pH 8.0) with 0.5 µM myosin and 3.8 mM NH$_4^+$-ATP as previously described (*Trybus, 2000*). Briefly, the amount of phosphate liberation as a function of time, determined with a colormetric method that detects phosphomolybdate, was determined by sampling the reaction every 2 min over a 10-min period. A standard curve with known phosphate concentrations was used to determine free phosphate concentration at each time point. The experiment was performed in triplicate with a single preparation of E525K and WT M2βS1 and 15HPZ.GFP.

### In vitro motility assays
The in vitro motility (IVM) assay was performed with WT and E525K M2β 15HPZ.GFP constructs, adapted from previously established protocols (*Kron et al., 1991*; *Rasicci et al., 2021*). Briefly, microscope cover slips were coated with 1% nitrocellulose in amyl acetate (Ladd Research) and applied to a microscope slide with double-sided tape to create a flow cell. Myosin in MOPS 20 buffer at concentrations between 72 and 90 µg mL$^{-1}$ (0.4–0.5 µM) was applied directly to the nitrocellulose surface, and the surface subsequently was blocked with BSA (1 mg mL$^{-1}$). To ensure inactive myosin heads ('dead heads') were blocked, unlabeled sheared actin (2 µM) was added to the flow cell and chased with ATP (2 mM). Actin was labeled with phalloidin-Alexa 555 (DsRed filter; excitation/emission 555/588 nm). To initiate motility, an activation buffer containing 0.35% methylcellulose, an ATP regeneration system (2 mM ATP, 5 mg mL$^{-1}$ glucose, 46 units mL$^{-1}$ pyruvate kinase, and 0.46 mM phosphoenolpyruvate), and oxygen scavengers (0.1 mg mL$^{-1}$ glucose oxidase and 0.018 mg mL$^{-1}$ catalase) was added to the flow cell. Temperature (25 ± 1°C) was monitored using a thermocouple meter (Stable Systems International). The slide was visualized promptly with a NIKON TE2000 microscope equipped with a 60×/1.4 NA phase objective and a Perfect Focus System. All images were acquired at 1 s intervals for 2 min using a shutter controlled CoolSNAP HQ2 cooled CCD digital camera (Photometrics) binned 2×2. Videos were exported to ImageJ and prepared for automated FAST software motility analysis (*Aksel et al., 2015*), from which >1000 actin filaments from one experiment (i.e. slide) per protein preparation (N=3) at 0.4–0.5 µM myosin were compiled for statistical analysis, WT vs. E525K.

### Transient kinetic measurements
An Applied Photophysics stopped-flow equipped with an excitation monochromator, 1.2 ms dead-time, 9.3 nm band pass, and temperature control (25°C) was used for all experiments. Fluorescence transients were fit to sum of exponentials using the stopped-flow program or GraphPad Prism. For example, the following function was used to fit fluorescence decays, $F(t) = F + \sum_{i=1}^{n} A_i e^{-k_i t}$, where $F(t)$ is the fluorescence as a function of time $t$, F$_\infty$ is the intensity at infinity, $A_i$ is the amplitude, $k_i$ is the observed rate constant characterized by the *ith* transition, and $n$ is the total number of observed transitions.

## Single turnover measurements

The mant fluorescence was examined with 290 nm excitation and a 395 nm long pass emission filter. Myosin samples were incubated on ice for 10 min in the appropriate buffer conditions prior to each experiment. Single mantATP turnover experiments were performed by incubating M2β constructs (0.25 µM) with mantATP (1 µM) for 30 s at room temperature and then mixing the complex with saturating ATP (2 mM) and monitoring the fluorescence decay over a 1000 s period. The average ( ± SD) SRX ratio, SRX rate, and DRX rate were determined from three separate protein preparations (one technical replicate per preparation).

## Steady-state FRET measurements in fluorimeter

We examined the fluorescence spectrum of M2β 15HPZ.GFP (0.25 µM) in the presence and absence of Cy3ATP (1 µM) at 25°C in a fluorimeter equipped with excitation and emission monochromators (Photon Technology International). The sample was excited at 470 nm, and the emission spectra were measured from 480 to 625 nm. The emission spectra of the donor were corrected for photobleaching and dilution effects, and the emission spectra of the acceptor were also correct for donor fluorescence to visualize the changes in the acceptor spectra.

## Stopped-flow FRET measurements

Stopped-flow FRET was examined by monitoring the change in donor fluorescence (GFP) using an excitation wavelength of 470 nm and measuring the emission with an interference filter (500–525 nm), which eliminated background fluorescence from Cy3ATP. For measurement of the rate of Cy3ATP binding to M2β, we mixed M2β 15HPZ.GFP with varying concentrations of Cy3ATP (0.25–2.5), pre-equilibrated in MOPS buffer containing 20 or 150 mM KCl (2–3 technical replicates per condition with a single protein preparation). For determining the FRET efficiency as a function of KCl concentration, M2β 15HPZ.GFP (0.25 µM) was mixed with Cy3ATP (1 µM), pre-equilibrated in MOPS 20 buffer with varying KCl concentrations. The fluorescence transients, average of three replicates, were fit to a single or double exponential function, and the total amplitude of the fluorescence change was utilized to determine the degree of donor quenching in each experiment. The steady-state FRET efficiency (*E*) was calculated by examining donor quenching, after correcting for background fluorescence, using the following equation (*Lakowicz, 2006*; *Tang et al., 2021*),

$$E \ = \ 1 - \frac{F_{DA}}{F_D}$$

where $F_{DA}$ is the donor fluorescence intensity in the presence of acceptor (under conditions of saturating Cy3ATP bound to M2β 15HPZ.GFP), and $F_D$ is the donor fluorescence intensity in the absence of acceptor (M2β 15HPZ.GFP in the presence of ATP). The entire series of FRET experiments at each KCL concentration was repeated with three separate protein preparations to determine the average FRET efficiency. The distance (*r*) between the donor and acceptor was calculated based on the equation below (*Lakowicz, 2006*),

$$r \ = \ R_0 \ \left[ \left(1 - E\right)/E \right]^{\frac{1}{6}}$$

where the Förster distance ($R_0$), the distance at which energy transfer is 50% efficient, was determined to be 63 Å (*Fessenden and Yuan, 2009*).

## Time resolved FRET measurements

Fluorescence lifetime measurements of M2β 15HPZ.GFP were performed using time-correlated single photon counting (TCSPC; DeltaPro, Horiba Scientific) with a 479 nm pulse diode laser and a 515 nm long-pass emission filter. We examined M2β constructs (60 nM) in the absence of nucleotide, presence of 0.1 µM Cy3ATP, and in the presence of saturating ATP (1 mM) and 0.1 µM Cy3ATP (ATP chase). The myosin samples were equilibrated in the appropriate buffer conditions for 10 min on ice, incubated with or without Cy3ATP for 1 min at room temperature, and measured in TCSPC. The TCSPC measurements took 30–45 s to complete. Then a second TCSPC measurement was taken after the addition of 1 mM ATP (ATP chase). The fluorescence lifetime decays were fit to two-exponential function and used to determine the average lifetime of the donor alone ($\tau_D$) and donor+acceptor ($\tau_{DA}$). The FRET efficiency was calculated using the following equation:

$$E = 1 - \frac{\tau_{DA}}{\tau_D}$$

Three technical replicates were performed on a single protein preparation, and the average FRET efficiency (± SD) was then used to determine the FRET distance (*r*) as described above.

## Electron microscopy
### Sample preparation and imaging
The samples (WT and E525K M2β 15HPZ.GFP) were diluted in 20 mM KCl, 10 mM MOPS, 1 mM EGTA, 0.5 mM ATP, 2 mM $MgCl_2$, and pH 7.4 to a final concentration of 77 nM and incubated at room temperature for 10 min. Negative staining was performed using 400 mesh copper grids with a thin carbon film on top. The grids were glow discharged for 45 s at 15 mA in a PELCO easiGlow to make the surface hydrophilic, aiding stain spreading. 5 µl of sample was applied to the grid for ~5 s to minimize disruption of the IHM due to binding to the carbon (*Burgess et al., 2007*), and the grid immediately rinsed with 1% (w/v) uranyl acetate for negative staining. Grids were imaged on a FEI Tecnai Spirit Transmission Electron Microscope at 120 kV with a Rio 9 3K × 3K CCD camera (Gatan). The EM experiments with WT and E525K were repeated in three separate protein preparations, each performed in parallel in order to compare the WT and mutant on grids prepared on the same day under identical conditions.

### Calculation of percentage folded molecules

A total of 4–9 micrographs (257–318 molecules) for each replicate (three separate protein preparations) of each construct were analyzed, and visual counting was performed to calculate the percent of folded molecules.

### 2-D classification and averaging

A total of 55 micrographs with 2.6 Å pixel size were used to extract 1397 (WT) and 1845 particles (E525K) in Relion 3.1.2 (*Scheres, 2012*). The particles were classified into 50 classes with a mask size of 520 Å, and the top 10 classes were used to make the comparison.

## Statistics
For this work, a biological replicate is defined as a new protein preparation, whereas a technical replicate is defined as a repeated experiment from the same stock of protein. Reported sample sizes refer specifically to biological replicates (except where noted), and a minimum of N=3 is reported for all experiments. Groups were allocated as WT vs. E525K or S1 vs. HMM, unless otherwise noted, and direct comparisons were made between these two groups. Accordingly, unpaired Student's *t*-tests were performed to compare differences between the groups in all experiments with statistical error. To avoid technical replicates in the IVM assay (i.e. actin filaments from the experiment), sample means of independent experiments were averaged, and the data have been presented as SuperPlots, as previously described (*Lord et al., 2020*; *Rasicci et al., 2021*). Stuck filaments (velocity = 0 nm s$^{-1}$) were excluded from this analysis. Otherwise, no data was excluded from any other reported experiment. The p-values for all statistical comparisons can be found in the source files.

## Acknowledgements
This work was supported by NIH grants to HL127699 to CMY, HL150953 to CMY and SS, AR072036, HL139883, HL164560 to RC, and an AHA Post-doctoral Fellowship to PT.

## Additional information

### Funding

| Funder | Grant reference number | Author |
|---|---|---|
| National Institutes of Health | HL127699 | Christopher M Yengo |
| National Institutes of Health | HL150953 | Christopher M Yengo Sivaraj Sivaramakrishnan |
| National Institutes of Health | AR072036 | Roger Craig |
| National Institutes of Health | HL139883 | Roger Craig |
| National Institutes of Health | HL164560 | Roger Craig |
| American Heart Association | 916818 | Prince Tiwari |

The funders had no role in study design, data collection and interpretation, or the decision to submit the work for publication.

### Author contributions

David V Rasicci, Data curation, Writing – review and editing, Formal analysis, Writing - original draft; Prince Tiwari, Fredrik R Sadler, Data curation, Writing – review and editing, Writing - original draft; Skylar ML Bodt, Data curation, Writing - original draft; Rohini Desetty, Data curation, Methodology, Writing - original draft; Sivaraj Sivaramakrishnan, Writing – review and editing, Supervision, Writing - original draft; Roger Craig, Conceptualization, Writing – review and editing, Supervision, Funding acquisition, Writing - original draft; Christopher M Yengo, Conceptualization, Data curation, Writing – review and editing, Funding acquisition, Formal analysis, Project administration, Writing - original draft

### Author ORCIDs

Sivaraj Sivaramakrishnan ⓘ http://orcid.org/0000-0002-9541-6994
Roger Craig ⓘ http://orcid.org/0000-0002-9707-5409
Christopher M Yengo ⓘ http://orcid.org/0000-0003-3987-9019

### Decision letter and Author response

Decision letter https://doi.org/10.7554/eLife.77415.sa1
Author response https://doi.org/10.7554/eLife.77415.sa2

## Additional files

### Supplementary files
• Transparent reporting form

### Data availability

All data generated or analysed during this study are included in the manuscript and supporting file; Source Data files have been provided for Figures 2-6.

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
