## [Editor Report]

This fundamental study demonstrates that a point mutation resulting in dilated cardiomyopathy in human cardiac myosin increases the fraction of molecules that adopt the auto-inhibited super-relaxed conformation. This provides a mechanism for the lower force output observed in the hearts of affected individuals. The data supporting this, utilizing kinetic methods, a FRET-biosensor to detect conformational changes, and electron microscopy are convincing.

---

## [Decision Letter]

**Decision letter after peer review:**

Thank you for submitting your article "Dilated cardiomyopathy mutation E525K in human β-cardiac myosin stabilizes the interacting heads motif and super-relaxed state of myosin" for consideration by *eLife*. Your article has been reviewed by 2 peer reviewers, and the evaluation has been overseen by a Reviewing Editor and Anna Akhmanova as the Senior Editor. The following individual involved in the review of your submission has agreed to reveal their identity: Kathleen M Ruppel (Reviewer #2).

Your manuscript has been reviewed by two expert referees and examined by myself. We all found the study timely and of great potential interest. However, both referees had substantial questions that must be addressed.

Essential revisions:

Referee #2 and myself were concerned about the low activity and poor quality of the movement of the 15 heptad construct. To rule out a substantial population of dead myosins, you should examine the high salt ATPase activity (K^+^ATPase). In addition, this reviewer also felt that you should do some characterization of the S1 construct bearing the mutation. Reviewer #1 felt that you did not deal sufficiently well with three published papers (Anderson 2018, Rohde 2018 and Chu 2021). All of us were concerned that the different techniques reported different ratios of SRX:DRX states (assuming that these correspond to the folded:unfolded conformations). I had a concern about Figure 4. I presume that the molecules are in the open conformation before the addition of the Cy3ATP. Given the magnitude of the highest rate constants that are measured in Figure 4 at 2.5 μm Cy3ATP, the folding must be fast if the data in E and F are reporting the rate of Cy3ATP binding and not the rate of folding. Again, fits, residuals and more information on the replications, etc are merited here. For example, what [Cy3ATP] was used in A and B? It must be the lowest values used looking at the amplitudes in A compared to the range of amplitudes in the higher nucleotide concentration ranges. What do the traces look like at high nucleotide where the rates should be much faster?

I am also appending the two reviews verbatim since each referee has numerous points for you to address.

*Reviewer 1 (Recommendations for the authors):*

1. There are no fluorescence spectra showing energy transfer.

2. It would be helpful to provide expected FRET distances for this pair.

3. In all cases where single and bi-exponentials are used it would be good to see the fits and residuals (supplementary would be fine).

4. Figure 4: I was very confused about what this assay was telling us, the fact you see a slower second-order binding constant would imply that the molecules are in the SRX/DRX combinations before nucleotide is applied, which would suggest that nucleotide-free myosin is in the IHM state. To my knowledge, IHM is stabilized by ADP.Pi and wouldn't exist without nucleotide. If the argument is that the signal follows the folding then that should occur at the rate constant for hydrolysis.

In addition, please explain why amplitudes of the whole combined transients are plotted instead of the individual processes. My thoughts would be that the two separate processes could have different affinities and therefore they should be plotted separately. Also, how well can the slow rate constants be determined with only a 30s acquisition? What are the errors and how many repeats have been made for these observations? These error values should be in the figure legend.

5. The discussion on using FRET (page 6 second to the last paragraph) was confusing (for me). The fluorescence lifetimes measured were stated to not have consistency, but then these values were used in Figure 5D to claim E525K was more folded than WT at higher salt. I wondered if the two lifetimes could actually represent the dynamics of an SRX-DRX transition? How would that pan out on the timescale of the experiment performed? How long were the samples incubated before measurement? And therefore, how much ATP was hydrolyzed?

6. Negative stain section (last paragraph page 6): What are the errors on these percentage values? How many repeats?

7. This statement: "The clinical significance of this study is that DCM mutations can stabilize the auto-inhibited state of myosin, providing a mechanistic basis for the reduced muscle force and power observed in DCM patients" seems to be an over-stretch from the results of a single mutation. The discussion has a more balanced view of this, stating that many DCM mutations affect other processes and DCM literature is less certain about the reduced activity argument.

8. Last section before the summary and future directions: Surely the fact that nearly 100% of HMM forms SRX at 20 mM KCl should suggest that the signal there represents a fully IHM molecule if the hypothesis is correct. The use of mava would also help here, mava is known to both slow down the ATPase and also allow the formation of IHM. There are conditions, where mava will push nearly everything into the SRX, if you don't see further formation of the FRET sensed IHM then that would suggest this isn't a great IHM sensor. Following the work of Anderson would really help here. They saw approx. 50% SRX without mava that went up to 100% at 25 mM Kac.

*Reviewer 2 (Recommendations for the authors):*

1. The author's claim that E525K myosin has a lower kcat value is based on ATPase data from 15-*hep* constructs which have unusually low activity and extremely high KATPase values (higher than the highest actin concentration used to generate these curves for WT protein). These not only raise doubts about the quality of the protein but will also result in erroneous extrapolated values. The authors should ideally use the S1 constructs to compare kcat values, they already have the viruses for those proteins. The authors should also provide gel images of the purified 15 *hep* proteins showing the three polypeptide chains.

2. It is not clear, and the authors do not discuss, what the fast and slow phases in the Cy3ATP-binding experiments correspond to. WT and E525K proteins have dramatically different proportions (65% vs. 25% of Afast respectively) of the two phases at low salt- a condition where they have similar SRX (STO assays) and IHM (FRET efficiency) proportions. What is the salt dependence of the rates and proportions of these two phases for WT and E525K proteins? It may help the authors to speculate about the origin of these two phases. It appears from panel 4B that these rates are different for E525K at low vs. high salt, but the authors don't comment on that.

3. The authors should discuss the limitation in the revised manuscript that the R0 value of the FRET pair used here doesn't allow precise measurement of the distances being probed here to be made and refrain from converting FRET efficiency values to distances.

4. While it is great that the FRET efficiency values calculated from TR-FRET analysis agree well with the values obtained from Cy3ATP binding experiments at low and high salt, it is not ideal to generate a salt-dependence of the FRET efficiency from TR-FRET experiments from just two data-points as shown in Figure 5D. It may be better to present that data as a comparison within 5A itself without drawing a line through them.

[Editors' note: further revisions were suggested prior to acceptance, as described below.]

Thank you for resubmitting your work entitled "Dilated cardiomyopathy mutation E525K in human β-cardiac myosin stabilizes the interacting heads motif and super-relaxed state of myosin" for further consideration by *eLife*. Your revised article has been evaluated by both of the original reviewers, Anna Akhmanova (Senior Editor) and a Reviewing Editor.

Both the original reviewers and the reviewing editor have read your rebuttal statement and the revised manuscript. All agree that it is much improved although there are still a few points that they would like you to consider. The main concern deals with the idea of 2 different apo-state 15HPZ populations ('open' and 'alt') that differ with respect to their ATP binding affinity. There are some cases in the myosin literature where two exponential nucleotide binding transients have been detected for various myosins, and often the idea of a slow equilibrium between a competent and an incompetent conformation of the myosins has been proposed as opposed to two static populations with inherently different binding rates (See Geeves et al. JBC 275:21624, 2000 for example). Have you considered this? The other concern is that ATP binding to apo-β-cardiac HMM has been reported in the literature to be fit to a single exponential using different experimental set-ups/detection methods:

a. Bovine cardiac HMM: Rohde et al. 2015, (doi: 10.1073/pnas.1514859112)

(TR)2 FRET, donor fluorophore is on the RLC, acceptor fluorophore Cy3-ATP

These authors report that "Cy3-ATP binding to labeled HMM induces a single-exponential time-dependent change in the total fluorescence of the Alexa Fluor donor, consistent with Cy3-ATP binding and formation of a high-FRET state." (Here the high FRET state is the pre-power stroke state.) Performed under low salt conditions.

b. Porcine cardiac HMM: Liu et al. (https://doi.org/10.1021/bi5015166)

Monitored intrinsic tryptophan fluorescence of porcine ventricular HMM at 25 mM KCl upon binding of ATP- data was fitted to a single exponential.

We realize your readout is different (at least in terms of the position of the donor fluorophore compared with Rohde), and that they are using human protein, but it still seems unlikely that these published studies would see no evidence of two populations under experimental conditions (WT, 20mM KCl) where you see Afast/Aslow of 65%/35%. Also, you are hypothesizing that the population with the decreased binding affinity may represent the IHM state, but then wouldn't you expect to have a smaller Afast for the E525K mutant at 150 mM KCl than WT given your other data?

Given these literature results, we think it is reasonable to ask that you examine the binding of Cy3ATP to the myosin constructs directly to complement the FRET experiments. Based on your new model you would expect to see a bi-exponential rate process.

---

## [Author Response]

Essential revisions:Referee #2 and myself were concerned about the low activity and poor quality of the movement of the 15 heptad construct. To rule out a substantial population of dead myosins, you should examine the high salt ATPase activity (K^+^ATPase).

We have performed NH_4_^+^ ATPase assays to demonstrate the number of active heads in our myosin preparations (Results-page 4, Methods – pages 11-12). We found that the high-salt ATPase activity was similar in S1 and 15HPZ.GFP demonstrating the 15HPZ.GFP construct does not disrupt folding of the myosin motor domain and that the number of active heads is similar in both preparations. We did find that the E525K mutation increased the high-salt ATPase 20% compared to WT in both the single and double-headed construct.

In addition, this reviewer also felt that you should do some characterization of the S1 construct bearing the mutation.

We now report the steady-state ATPase activity for the E525K M2β S1 construct (Page 4 and Figure 2). We demonstrate, surprisingly, that this mutation enhances actin-activated ATPase activity by increasing the k_cat_ and reducing the K_ATPase_. We feel the E525K S1 results are extremely interesting since they demonstrate that the mutation is capable of stabilizing the SRX state even though the motor domain has enhanced intrinsic ATPase activity (see discussion on page 10). We did not perform in vitro motility with the E525K S1 construct since we felt this was another question that is tangential to the main focus of the current paper. Mainly, the question of how E525K specifically alters the intrinsic motor and ATPase activity will be reported in a separate paper that can provide mechanistic details on how the mutation alters myosin motor domain function.

Reviewer #1 felt that you did not deal sufficiently well with three published papers (Anderson 2018, Rohde 2018 and Chu 2021).

We have enhanced our discussion of the Anderson, Rohde, and Chu papers in the discussion (see page 8).

All of us were concerned that the different techniques reported different ratios of SRX:DRX states (assuming that these correspond to the folded:unfolded conformations).

The results of the single turnover and IHM FRET are fairly consistent in terms of the KCl dependance of the SRX:DRX and IHM:Open ratios (pages 4-6 and Figures 3&5). We agree that the EM, although it follows the same trend as the single turnover and FRET, has quite different fractions in the IHM than we observe in solution. We have discussed that this is likely caused by the interaction of the HMM with the EM grid surface, which disrupts the IHM conformation (see pages 6-7). To further address the question of the degree of variability in the EM experiments we have compared three separate preparations and examined the fractional amounts of IHM and open conformations in the WT and mutant (pages 6-7, Figure 6). We suggest surface interactions are known to affect protein conformation and are common in approaches that require attaching a protein to a surface (e.g. EM, AFM). We attempted to minimize these effects by minimizing the contact time of HMM with the grid before staining.

I had a concern about Figure 4. I presume that the molecules are in the open conformation before the addition of the Cy3ATP. Given the magnitude of the highest rate constants that are measured in Figure 4 at 2.5 μm Cy3ATP, the folding must be fast if the data in E and F are reporting the rate of Cy3ATP binding and not the rate of folding. Again, fits, residuals and more information on the replications, etc are merited here. For example, what [Cy3ATP] was used in A and B? It must be the lowest values used looking at the amplitudes in A compared to the range of amplitudes in the higher nucleotide concentration ranges. What do the traces look like at high nucleotide where the rates should be much faster?

We have added more detail to Figure 4. We now include the residuals of the fits in panels A and B. We added the concentration of Cy3ATP (1 µM) and M2β (0.25 µM) to the figure legend. We updated panels A and B by labelling the y-axis to fluorescence, instead of normalized fluorescence. We believe this generated confusion in the previous figure when comparing the y-axis in panels A and B with amplitudes in panels C and D. We did use 1 µM Cy3ATP final for these experiments which is a concentration that achieved saturation of the FRET amplitude (see panels C and D).

As requested by the reviewers, we enhanced the discussion of the Cy3ATP binding rate constants (page 5 and Figure 4—figure supplements 3 and 4). We agree that the folding into the IHM is relatively rapid following Cy3ATP binding. We have fit the fluorescence transients to a kinetic scheme in which ATP binding is followed by a transition into the IHM, which is rate-limited by the rate of ATP binding under the conditions of our experiments. We also find that since both the fast phase and the slow phase are dependent on ATP concentration, our data fits best to two conformations of myosin in the absence of nucleotide. We propose that one conformation is the open conformation, which binds ATP similar to single headed S1, and the second conformation is an alternative conformation that binds ATP more slowly. We speculate that the alternative conformation could be the IHM or another conformation of the motor domain, however further study is necessary to examine this question (see Discussion page 9).

Reviewer 1 (Recommendations for the authors):1. There are no fluorescence spectra showing energy transfer.

We added a figure demonstrating the fluorescence spectra of the donor in the presence and absence of acceptor (see Figure 4—figure supplement 1).

2. It would be helpful to provide expected FRET distances for this pair.

We provide the expected FRET distances in Figure 1 and in page 3 of the results.

3. In all cases where single and bi-exponentials are used it would be good to see the fits and residuals (supplementary would be fine).

See fits and residuals now for all Figures (Figure 3—figure supplements 1 and 2; Figure 4, Figure 5—figure supplement 1).

4. Figure 4: I was very confused about what this assay was telling us, the fact you see a slower second-order binding constant would imply that the molecules are in the SRX/DRX combinations before nucleotide is applied, which would suggest that nucleotide-free myosin is in the IHM state. To my knowledge, IHM is stabilized by ADP.Pi and wouldn't exist without nucleotide. If the argument is that the signal follows the folding then that should occur at the rate constant for hydrolysis.

Our model suggests there are two conformations of nucleotide-free HMM before mixing with Cy3ATP, which gives rise to the fast and slow ATP concentration-dependent rate constants. We suggest the fast rate constant, which is similar to S1, is the open state and the slow component is an alternate conformation. Our model also suggests that the folding occurs fast relative to the ATP binding step. It is true that the alternate conformation could be the IHM even though previous work predicts that the IHM would only form after ATP binding and hydrolysis. Therefore, we suggest further study is necessary to examine the possibility that the IHM may form in nucleotide-free conditions (see pages 5, 8-9).

In addition, please explain why amplitudes of the whole combined transients are plotted instead of the individual processes. My thoughts would be that the two separate processes could have different affinities and therefore they should be plotted separately. Also, how well can the slow rate constants be determined with only a 30s acquisition? What are the errors and how many repeats have been made for these observations? These error values should be in the figure legend.

The amplitudes of the slow and fast components were combined to obtain the overall formation of the IHM conformation in low and high salt conditions. We feel total amplitude is the most informative in comparing the low and high salt conditions. We now also report the amplitudes of both phases in low salt and high salt conditions (see Figure 4—figure supplement 2). We added that each data point represents 3 repetitions from 1 preparation and the errors represent standard error of the fit. A summary of the rate constants and errors is given in Table 4.

5. The discussion on using FRET (page 6 second to the last paragraph) was confusing (for me). The fluorescence lifetimes measured were stated to not have consistency, but then these values were used in Figure 5D to claim E525K was more folded than WT at higher salt. I wondered if the two lifetimes could actually represent the dynamics of an SRX-DRX transition? How would that pan out on the timescale of the experiment performed? How long were the samples incubated before measurement? And therefore, how much ATP was hydrolyzed?

We removed the confusing statement about consistent differences in the short and long components of the lifetime decay, since this comment is relevant to distance distribution analysis which was not performed in this study. The myosin samples were equilibrated in the appropriate buffer conditions for 10 minutes on ice, incubated with Cy3ATP for one minute at room temperature, and measured by TCSPC. The TCSPC measurements were performed within a 30-45 second period (see Methods, page 13). We estimate that most of the myosin would be in the ADP.Pi state during the measurements.

Since the lifetimes are on a nanosecond timescale any structural changes on the millisecond and microsecond timescale can be observed by performing transient time resolved FRET (TR2-FRET), which requires specialized instrumentation. In the equilibrium experiments performed in this study we assume there is a distribution of open and IHM structural states and our results will be an ensemble average of the structural states populated in solution.

6. Negative stain section (last paragraph page 6): What are the errors on these percentage values? How many repeats?

We performed EM experiments on a total of 3 different preparations of WT and E525K enabling us to make 3 side-by-side comparisons with identical grid conditions in each case. We updated Figure 6 (pages 6-7) to demonstrate the similar trend in the data in all 3 preparations, in terms of the folded versus unfolded conformations detected. The absolute fraction of HMM molecules in the folded/unfolded states did vary from prep to prep, which we believe is due to variations in surface interactions with the EM grid from one experiment to the next. However, the ratio of folded WT/folded E525K was similar in each prep which we feel is the best comparison to make. We found a consistent 4-fold increase in the number of folded molecules in the mutant compared to the WT 15HPZ.GFP (Figure 6 and Table 5).

7. This statement: "The clinical significance of this study is that DCM mutations can stabilize the auto-inhibited state of myosin, providing a mechanistic basis for the reduced muscle force and power observed in DCM patients" seems to be an over-stretch from the results of a single mutation. The discussion has a more balanced view of this, stating that many DCM mutations affect other processes and DCM literature is less certain about the reduced activity argument.

We agree and thank the reviewer for pointing this out. We now state that the manuscript demonstrates that the stabilization of the SRX state is possible in DCM, while studying more mutations is crucial to reveal if this is a common mechanism in DCM (see page 7).

8. Last section before the summary and future directions: Surely the fact that nearly 100% of HMM forms SRX at 20 mM KCl should suggest that the signal there represents a fully IHM molecule if the hypothesis is correct. The use of mava would also help here, mava is known to both slow down the ATPase and also allow the formation of IHM. There are conditions, where mava will push nearly everything into the SRX, if you don't see further formation of the FRET sensed IHM then that would suggest this isn't a great IHM sensor. Following the work of Anderson would really help here. They saw approx. 50% SRX without mava that went up to 100% at 25 mM Kac.

We thank the reviewer for this exciting suggestion. We also thought of the mavacamten experiment, but we ultimately decided this would be best performed in a new study that can fully examine the impact of this drug on 15HP HMM in varying ionic strength conditions. In addition, to thoroughly study the impact of mavacamten we would need to examine the FRET as a function of Cy3ATP concentration at low and high salt in several preparations. In addition, if the drug alters Cy3ATP binding the FRET biosensor would be greatly impacted.

Reviewer 2 (Recommendations for the authors):1. The author's claim that E525K myosin has a lower kcat value is based on ATPase data from 15-hep constructs which have unusually low activity and extremely high KATPase values (higher than the highest actin concentration used to generate these curves for WT protein). These not only raise doubts about the quality of the protein but will also result in erroneous extrapolated values. The authors should ideally use the S1 constructs to compare kcat values, they already have the viruses for those proteins. The authors should also provide gel images of the purified 15 hep proteins showing the three polypeptide chains.

The S1 experiments have been added to the manuscript as described above (see Figure 2B and Table 1). The gel in supplemental data shows the comparison of the S1 and 15HPZ constructs (see Figure 2—figure supplement 1).

2. It is not clear, and the authors do not discuss, what the fast and slow phases in the Cy3ATP-binding experiments correspond to. WT and E525K proteins have dramatically different proportions (65% vs. 25% of Afast respectively) of the two phases at low salt- a condition where they have similar SRX (STO assays) and IHM (FRET efficiency) proportions. What is the salt dependence of the rates and proportions of these two phases for WT and E525K proteins? It may help the authors to speculate about the origin of these two phases. It appears from panel 4B that these rates are different for E525K at low vs. high salt, but the authors don't comment on that.

We have added more details to the Cy3ATP binding FRET results and now include a kinetic model to interpret our FRET transients (see pages 5, 9, Figure 4—figure supplements 3&4). See description above.

3. The authors should discuss the limitation in the revised manuscript that the R0 value of the FRET pair used here doesn't allow precise measurement of the distances being probed here to be made and refrain from converting FRET efficiency values to distances.

We now point out the limitations of the FRET based on the R_0_ and efficiencies measured (see pages 6 and 10).

4. While it is great that the FRET efficiency values calculated from TR-FRET analysis agree well with the values obtained from Cy3ATP binding experiments at low and high salt, it is not ideal to generate a salt-dependence of the FRET efficiency from TR-FRET experiments from just two data-points as shown in Figure 5D. It may be better to present that data as a comparison within 5A itself without drawing a line through them.

We have removed the line in the TR-FRET graph (Figure 5D).

[Editors' note: further revisions were suggested prior to acceptance, as described below.]

Both the original reviewers and the reviewing editor have read your rebuttal statement and the revised manuscript. All agree that it is much improved although there are still a few points that they would like you to consider. The main concern deals with the idea of 2 different apo-state 15HPZ populations ('open' and 'alt') that differ with respect to their ATP binding affinity. There are some cases in the myosin literature where two exponential nucleotide binding transients have been detected for various myosins, and often the idea of a slow equilibrium between a competent and an incompetent conformation of the myosins has been proposed as opposed to two static populations with inherently different binding rates (See Geeves et al. JBC 275:21624, 2000 for example). Have you considered this? The other concern is that ATP binding to apo-β-cardiac HMM has been reported in the literature to be fit to a single exponential using different experimental set-ups/detection methods:a. Bovine cardiac HMM: Rohde et al. 2015, (doi: 10.1073/pnas.1514859112)(TR)2 FRET, donor fluorophore is on the RLC, acceptor fluorophore Cy3-ATPThese authors report that "Cy3-ATP binding to labeled HMM induces a single-exponential time-dependent change in the total fluorescence of the Alexa Fluor donor, consistent with Cy3-ATP binding and formation of a high-FRET state." (Here the high FRET state is the pre-power stroke state.) Performed under low salt conditions.b. Porcine cardiac HMM: Liu et al. (https://doi.org/10.1021/bi5015166)Monitored intrinsic tryptophan fluorescence of porcine ventricular HMM at 25 mM KCl upon binding of ATP- data was fitted to a single exponential.We realize your readout is different (at least in terms of the position of the donor fluorophore compared with Rohde), and that they are using human protein, but it still seems unlikely that these published studies would see no evidence of two populations under experimental conditions (WT, 20mM KCl) where you see Afast/Aslow of 65%/35%. Also, you are hypothesizing that the population with the decreased binding affinity may represent the IHM state, but then wouldn't you expect to have a smaller Afast for the E525K mutant at 150 mM KCl than WT given your other data?

The Afast was the dominant signal at 150 mM KCl for WT (0.81) and E525K (0.89). We hypothesize that both WT and E525K are mostly fully open in the Apo state at high salt. Upon ATP binding WT does not favorably transition into the IHM while E525K is still capable of making this transition (see revised section of the Results – page 5-6). Also, we revised the following sentence in the Discussion to clarify this point “We also found that high salt shifts the equilibrium toward the Open conformation in the absence of nucleotide for both WT and E525K, while only the mutant can transition efficiently into the IHM upon ATP binding”. Also, in the discussion we removed the reference to the suggestion that the alternate or incompetent state could be the IHM (see Discussion – page 9).

Given these literature results, we think it is reasonable to ask that you examine the binding of Cy3ATP to the myosin constructs directly to complement the FRET experiments. Based on your new model you would expect to see a bi-exponential rate process.

We measured the Cy3ATP binding as suggested by the reviewers. We found that the fluorescence transients were best fit to a three-exponential function, with two phases that are dependent on Cy3ATP concentration and a third phase that is independent. The third nucleotide-independent slow phase could be an isomerization of the Cy3 fluorophore in the active site. The two different Cy3ATP isoforms have been shown to have distinct fluorescence signatures, which can complicate interpretations of the fluorescence transients of the mixed isomer (Oiwa et al. 2003, Biophys J. 84: 634-642). Since we are monitoring FRET by donor quenching (GFP) the slow phases in the Cy3 fluorescence does not contribute to the FRET signal. Overall, we do observe two rate constants that are linearly dependent on Cy3ATP concentration as we do in the FRET studies.

The direct binding results are in agreement with our model that suggests there are two static states with different ATP binding kinetics. However, after looking more closely at the Geeves model, we conclude that the direct binding data does not rule out the possibility that there are two states in the absence of nucleotide that are in equilibrium, one that is competent to bind nucleotide and another incompetent. At this range of Cy3ATP concentrations (0.25-2.5 µM) the Geeves model would likely observe two rates that are linearly dependent on Cy3ATP concentration . Therefore, we are in favor of not including the direct Cy3ATP binding measurements because they do not provide additional evidence about the kinetic mechanism. Also, the results are a distraction from the main theme of the paper which is focused on the correlation between the SRX and IHM states and the impact of the E525K mutation. In order to explain the direct binding results it would require significant space and an additional supplementary Figure. Also, there is the issue of the 3rd exponential that is not defined.

Therefore, we have revised the paragraph in the Results (page 5-6) and in the Discussion (see page 9) to discuss the other possible kinetic mechanism (competent vs incompetent states in equilibrium). We include diagrams of both models in the supplemental data for clarity (see Figure 4—figure supplement 3).